# The INALT* family - a set of high-resolution nests for the Agulhas Current system within global NEMO ocean/sea-ice configurations

Franziska U. Schwarzkopf[1], Arne Biastoch[1,2], Claus W. Böning[1,2], Jérôme Chanut[3], Jonathan V. Durgadoo[1], Klaus Getzlaff[1], Jan Harlaß[1], Jan K. Rieck[1], Christina Roth[1], Markus M. Scheinert[1], and René Schubert[1]

[1]GEOMAR Helmholtz Centre for Ocean Research Kiel, Düsternbrooker Weg 20, 24105 Kiel, Germany
[2]Kiel University, Christian-Albrechts-Platz 4, 24118 Kiel, Germany
[3]Mercator-Ocean, Parc Technologique du Canal, 8-10 rue Hermès, 31520 Ramonville St-Agne, France

**Correspondence:** Franziska U. Schwarzkopf (fschwarzkopf@geomar.de)

**Abstract.** The Agulhas Current, the western boundary current of the South Indian Ocean, has been shown to play an important role in the connectivity between the Indian and Atlantic oceans. The greater Agulhas Current system is highly dominated by mesoscale dynamics. To investigate their influence onto the regional and global circulations, a family of high-resolution ocean general circulation model configurations based on the NEMO code has been developed. Horizontal resolution refinement is achieved by embedding "nests" covering the South Atlantic and the western Indian oceans at 1/10° (INALT10) and 1/20° (INALT20) within global hosts with coarser resolutions. Nests and hosts are connected through two-way interaction, allowing the nests not only to receive boundary conditions from their respective host, but also to feed back the impact of regional dynamics onto the global ocean. A double-nested configuration at 1/60° resolution (INALT60) has been developed to gain insights into sub-mesoscale processes within the Agulhas Current system. Large-scale measures such as the Drake Passage transport and the strength of the Atlantic meridional overturning circulation are rather robust among the different configurations indicating the important role of the hosts in providing a consistent embedment of the regionally refined grids into the global circulation. The dynamics of the Agulhas Current system strongly depend on the representation of mesoscale processes. Both, the southward flowing Agulhas Current and the northward flowing Agulhas Undercurrent increase in strength with increasing resolution towards more realistic values, which suggests the importance of improving mesoscale dynamics as well as bathymetric slopes along this narrow western boundary current regime. The exploration of numerical choices such as lateral boundary conditions and details of the implementation of surface wind stress forcing demonstrates the range of solutions within any given configuration.

\* The name INALT was introduced by Durgadoo et al. (2013) as an acronym for "inaliti" meaning needle in isiXhosa. "Agulhas" is Portugese for needles. INALT can also be read as "INdian-AtLanTic".

# 1 Introduction

The waters around southern Africa are an important player in the world-wide system of ocean currents. The Agulhas Current (AC), one of the strongest currents in the world ocean, is the western boundary current in the South Indian Ocean and has a far-reaching influence into the Atlantic (Gordon, 2003; Lutjeharms, 2006). Forced by trade winds, it transports warm and saline water from the tropical Indian Ocean along the African coast towards the southern tip of Africa, where it performs an abrupt turn back into the Indian Ocean. Owing to the termination of the African continent at around 35° S and a range of non-linear processes, a part of the AC finds its way into the South Atlantic. It thus provides an important link for the surface limb of the global overturning circulation from the Pacific and Indian oceans into the Atlantic Ocean (Durgadoo et al., 2017; Le Bars et al., 2013; Beal et al., 2011). Modelling the AC system and its embedment within the gobal circulation demands for global ocean general circulation models with mesoscale resolution in the region around southern Africa (e.g. Biastoch et al., 2008a; Holton et al., 2017)

The northern part of the AC (27° S to 34° S) flows relatively stable and close to the continental slope (Lutjeharms, 2006). Further south at around 34-35° S the shelf begins to widen and allows the AC to meander. Due to inertia, the current overshoots the African continental edge at 20° E before it abruptly retroflects back into the Indian Ocean. There it continues as the Agulhas Return Current and closes the subtropical gyre (Lutjeharms and Ansorge, 1997). The dynamics of the retroflection are an interplay between the wind forcing, inertia, bathymetry and non-linear dynamics (see box 1 in Beal et al., 2011). At the retroflection itself mesoscale eddies are shed (Pichevin et al., 1999; van Leeuwen et al., 2000) that interact, merge and split in the Cape Basin (Boebel et al., 2003; Laxenaire et al., 2018). With diameters of up to 500 km (Arhan et al., 1999) and average depth extensions of 2000 m, eventually reaching down to the bottom (Van Aken et al., 2003), Agulhas rings are among the largest mesoscale features in the world ocean, transporting large amounts of warm and saline Indian Ocean water into the relatively colder and fresher South Atlantic (Gordon, 1986).

Upstream the AC, in the Mozambique Channel and southeast of Madagascar, mesoscale eddies are also formed through local instabilities (Swart et al., 2010). These propagate into the western boundary current system, causing the AC to sometimes meander off-shore (De Ruijter et al., 1999). These so-called Natal Pulses exhibit timescales of 70 to 90 days and rapidly propagate downstream. They imprint a range of spatio-temporal scales on the AC that may impact the generation and fate of Agulhas rings (Rouault and Penven, 2011; Schouten et al., 2002).

Owing to the open setting in the Atlantic, Indian and Southern oceans, the AC system forms a bridge between the wind-driven circulation in the individual basins. It effectively is a key link in the southern hemisphere supergyre, combining both subtropical gyres of the Indian and the Atlantic oceans (Speich et al., 2007). The amount of Indian Ocean water that flows through the AC system and becomes part of the Atlantic circulation is termed Agulhas leakage (Gordon, 2003). A portion of this finds its way into the upper limb of the Atlantic Meridional Overturning Circulation (AMOC) (Rühs et al., 2013). The continuous inflow of Agulhas leakage influences the hydrography and watermass distribution in the Atlantic Ocean (Lee et al., 2011; Lübbecke et al., 2015; Biastoch et al., 2015). The amount of Agulhas leakage (and hence the influence on the Atlantic Ocean) is linked to atmospheric processes, more specifically the Southern Hemisphere Westerlies on decadal timescales (Durgadoo et al., 2013).

As part of a decadal upward swing, the Agulhas leakage was subject to a 30 % increase from the 1960s to the 2000s (Biastoch et al., 2009b, 2015).

The role of the AC system in the surface limb of the overturning circulation, is mirrored at depth by a flow of deep water masses through the AC system. Arhan et al. (2003) showed that around 11 Sv of North Atlantic Deepwater finds its way around the southern tip of Africa from west to east. While most of it is subject to a sluggish flow into the deep basins east and west of the Madagascar Ridge, it is also found in a northward flowing core beneath and in-shore of the AC (Casal et al., 2006). This Agulhas Undercurrent (AUC) is also subject to strong variability as a consequence of the interplay between Natal Pulses and the western boundary current regime (Beal, 2009; Biastoch et al., 2009a).

Model simulations have been increasingly successful in capturing the AC system over the past decades. In models with relatively coarse resolution below the first baroclinic Rossby radius (Chelton et al., 1998), the AC follows Sverdrupian dynamics and is able to perform its retroflection back into the Indian Ocean (de Ruijter et al., 1999). Agulhas rings begin to appear at resolutions of 1/3° to 1/4°, however too regular in size and pathways. Resolutions of about 1/10°, representing the Rossby radius of deformation in this region, are required to simulate the instability processes in the source regions of the AC as well as along the AC path (Natal Pulses) and to provide the correct current structure of the western boundary current, including the AUC (Biastoch et al., 2009a). It still remains to be explored whether a horizontal resolution of 1/10° is already sufficient to simulate the full range of mesoscale processes and whether integral numbers like Agulhas leakage depend on the previously unrepresented dynamics. This is in particular the case once sub-mesoscale dynamics are included in the simulation as has been shown for the Gulf Stream and the successional North Atlantic Current by Chassignet et al. (2017).

Despite locally resolving mesoscale dynamics, as can be achieved by regional models (e.g. Penven et al., 2006; Hermes et al., 2007), capturing Southern Ocean dynamics, that have an impact on the Agulhas leakage (Durgadoo et al., 2013) might also be necessary to realistically simulate the dynamics of the AC system. To study the impact of the large-scale circulation on the AC or the impact of the AC system on the global overturning, global models are required. However, the numerical costs of global mesoscale resolving models are still too high to perform several multi-decadal experiments. To overcome this conflict, nesting has been shown to provide a good compromise, allowing for regional high resolution and keeping the global context (Biastoch et al., 2018).

To address the above mentioned questions, a systematic hierarchy of model configurations based on the NEMO code, ranging from eddy-poor to eddy-rich configurations has been developed and is described here. Therein, the flagship configuration INALT20 (Fig. 1) covers the South Atlantic and the western Indian oceans, including the AC system and a Southern Ocean sector at mesoscale resolution (1/20°). Configurations at coarser, 1/10°, resolution, INALT10 and INALT10x, serve three purposes: they provide the comparison with previous configurations, the possibility to couple to an active atmosphere and enable the systematic evaluation to explore the influence of Southern Ocean dynamics. Un-nested versions of the host configurations at 1/2° (ORCA05) and 1/4° (ORCA025) provide a comparison of the high-resolution configurations to coarser global ocean models, e.g. being performed for longer timescales or in coupled simulations. Additionally, on the other end of the range in resolution, a secondary nest at 1/60°, INALT60, allows to explore the role of sub-mesoscale dynamics in the AC system. With this set of model configurations resolution convergence within the Agulhas system is investigated.

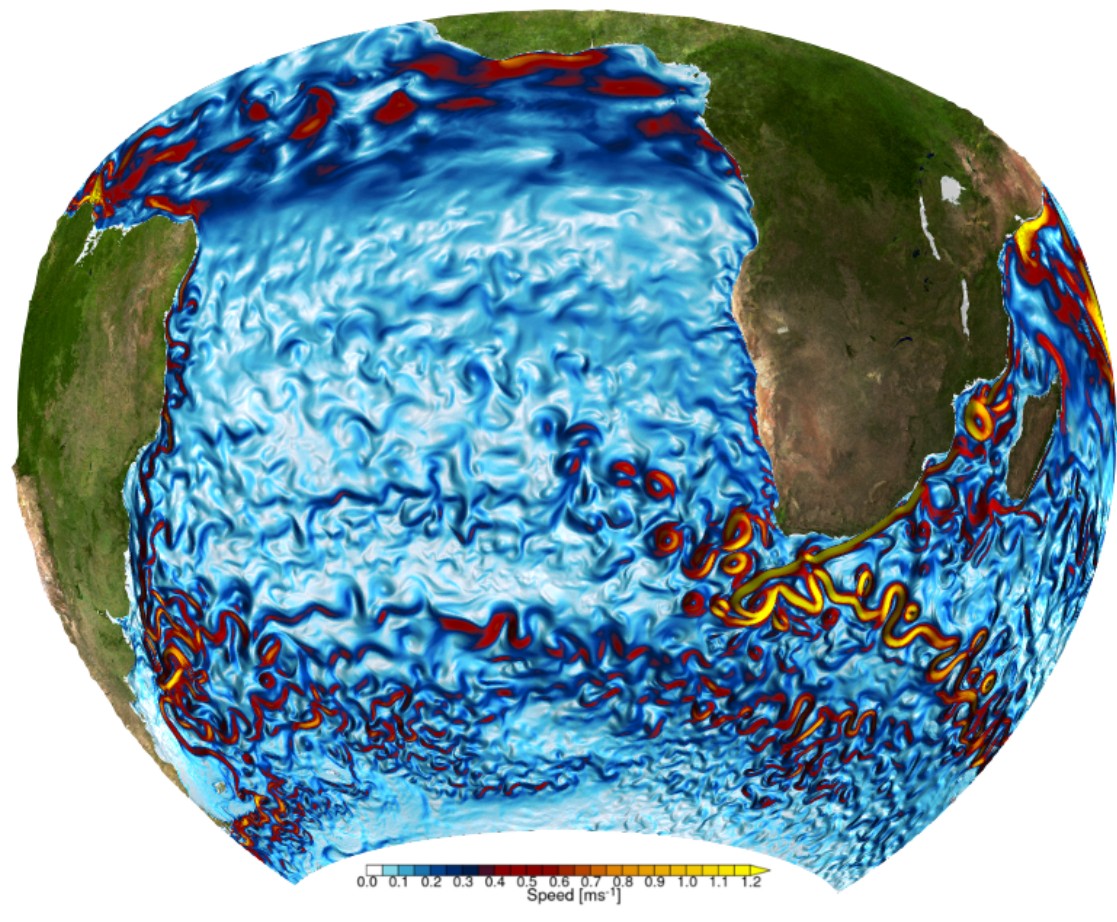

**Figure 1.** Snapshot of surface speed [m s$^{-1}$] as simulated in the nested area of INALT20.

## 2   The INALT family

The INALT family is a set of global ocean model configurations covering the greater AC system at high resolution. The predecessor of these configurations is INALT01 (Durgadoo et al., 2013), a well established 1/10° model configuration. INALT01 has been widely used to understand AC dynamics (Rühs et al., 2017; Malan et al., 2018), the effect of the Southern Hemisphere wind systems on the AC (Durgadoo et al., 2013; Loveday et al., 2014), the impact of Agulhas leakage on the Atlantic circulation and hydrography (Biastoch et al., 2015; Lübbecke et al., 2015), as well as local phenomena in the (greater) Agulhas region (Cronin et al., 2013; Malan et al., 2019) and has been utilized for a range of interdisciplinary applications (Scussolini et al., 2013; Steinhardt et al., 2014; van Sebille et al., 2015). The set of configurations described here updates INALT01 with respect to the code version. Additionally, the nested region is extended into the Southern Ocean and a range of resolutions is covered.

## 2.1 Configurations

The configurations introduced in this study are based on the ocean general circulation model NEMO (Nucleus for European Modelling of the Ocean code version 3.6, Madec and the NEMO team (2014)) coupled to the Louvain-La-Neuve sea-ice Model version 2 using a viscous-plastic rheology (LIM2-VP, Fichefet and Maqueda, 1997). Global configurations with tri-polar Arakawa-C grids, named ORCA, are used as a basis to build regionally finer resolved configurations realized by the AGRIF (Adaptive Grid Refinement in Fortran) library (Debreu et al., 2008). Hereby, a global, relatively coarsely resolved grid (hereafter: "host") is combined with a regionally confined, high-resolution grid (hereafter: "nest") allowing for two-way interactions: the host not only provides boundary conditions for the nest but also receives information from the nest. AGRIF is a very cost-effective technology to regionally refine the horizontal grid, with typically just 10-20% computational overhead through the global host (Biastoch et al., 2018), depending on the size of the nested region, the bigger, the less overhead.

The model solutions on the host and nest grids are integrated sequentially. Once the model is advanced by one time step on the host grid, the solution of two consecutive time steps are temporally and spatially interpolated to the nest grid and the corresponding finer time steps. Given these boundary conditions, the model is advanced by several time steps on the nest grid until the same time as on the host grid is reached. Afterwards, the barotropic solution (i.e. sea level and barotropic fluxes) on the host grid is updated with the solution from the nest grid along its boundaries. This is repeated for a certain number of time steps after which the solution on the host grid is updated with the solution on the nest grid everywhere inside the refined domain (baroclinic update, Debreu et al., 2008).

The flagship configuration INALT20 consists of a host at 1/4° horizontal resolution and an embedded nest at 1/20° resolution (Fig. 1), covering the South Atlantic and the western Indian oceans between 70° W and 70° E and from the northern tip of the Antarctic Peninsula at 63° S to 10° N (Fig. 2). Analogously, INALT10x covers the same nested region, but at 1/10° and embedded within a host at 1/2°. With the southern boundary located further north, at 50° S, INALT10 covers the same nest area as the predecessor INALT01 (Durgadoo et al., 2013). INALT10, on the one hand allows to directly compare the old and the new version of this configuration, and on the other hand allows to elucidate the influence of mesoscale dynamics in the Atlantic portion of the Antarctic Circumpolar Current (ACC) by direct comparison with INALT10x, in which this region is represented at eddying resolution. To shed light on sub-mesoscale processes around the southern tip of the African continent, INALT60 is introduced with a secondary 1/60° nest (20° W to 70° E and 50° S to 6.5° S), embedded within a spatially reduced 1/20° nest version INALT20r (20° W to 70° E and 50° S to 6.5° S). The sizes of the different configurations and computational costs required to simulate one model year are given in Table 1. In INALT10 and INALT60, the sea-ice model is only performed on the host grids, as ice does not enter the nested domain, while in INALT10x and INALT20 sea-ice is also represented on the refined grid. Additionally, the global grids ORCA025 (1/4° horizontal resolution, Barnier et al., 2006) and ORCA05 (1/2° horizontal resolution, Biastoch et al., 2008b) also exist as un-nested versions and are used to decipher the influence of the mesoscale processes onto large scale ocean dynamics by isolating the nest effect outside the nest regions (e.g., Biastoch et al., 2008a).

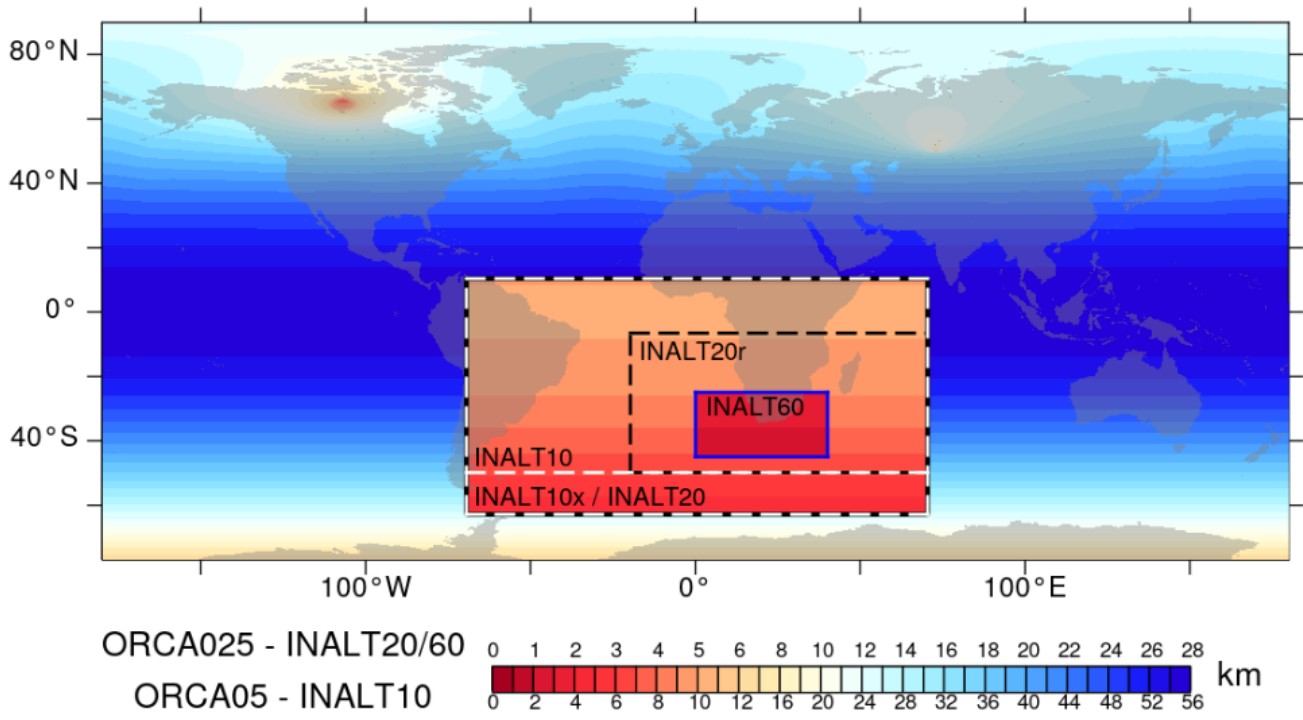

**Figure 2.** Grid sizes in the INALT family: shown is the zonal grid length [km] for the global ORCA025 grid with the embedded INALT20 (outer black box), INALT20r (inner black box) and INALT60 (blue box) grids (upper scale) as well as for ORCA05 with the embedded INALT10x (white dashed box, same as INALT20) and INALT10 with its southern boundary shifted north (lower scale).

The horizontal resolutions of these configurations for the greater AC system translate into grid sizes of 40 km and 20 km within ORCA05 and ORCA025, respectively, via 8 km and 4 km within INALT10(x) and INALT20(r) respectively, to less than 2 km within INALT60 (Fig. 2). In the eddy-poor configuration ORCA05, mesoscale effects are parameterized following Gent and McWilliams (1990). ORCA025 is eddy-active since it resolves the first baroclinic Rossby radius at mid-latitudes (Chelton et al., 1998). The grid sizes of the nested configurations are below this radius of 10 km at ∼55° S South of Africa and can therefore be referred to as "eddy-rich", representing the mesoscale.

All configurations share the same vertical grid with 46 z-levels varying in layer thickness from 6 m at the surface to 250 m in the deepest layers, resolving the first baroclinic mode (Stewart et al., 2017) which is needed for the representation of the mayor baroclinic currents. The same vertical grid has proven to be an appropriate choice for simulations with model configurations up to 1/20° horizontal resolution (e.g. Böning et al., 2016; Behrens et al., 2017). The bottom topography is represented by partial steps (Barnier et al., 2006) with a minimum layer thickness of 25 m. The bathymetry datasets in ORCA05 and ORCA025 have been developed within the DRAKKAR community and are described in Molines et al. (2006) and Barnier et al. (2006), respectively. For the nest grids at 1/20° and 1/60° resolution, bathymetries are interpolated from ETOPO1 and from ETOPO2 for the grids at 1/10° using the nesting tools (Lemarié, 2006).

**Table 1.** Grid sizes, timesteps for hosts (h) and nests (n1) as well as the secondary nest in INALT60 (n2), and grid point integrations per model day (GPIPD) are given for the individual configurations. The computational requirements in nodes consisting of 24 CPUs each, for the model code itself (NEMO) and the input-output server (XIOS), and CPU hours (per model year) are estimated on a Cray XC40, equipped with Intel Xeon Haswell processors, at the North-German Supercomputing Alliance ('Norddeutscher Verbund für Hoch- und Höchstleistungsrechnen'; HLRN).

| Configuration | horizontal dimension | timestep (h/n1/n2) | GPIPD | nodes (NEMO/XIOS) | CPU hours |
|---|---|---|---|---|---|
| ORCA05 | $722 \times 511$ | 2160 | $6.8 \times 10^8$ | 26 (25/1) | 300 |
| ORCA025 | $1442 \times 1021$ | 1440 | $41 \times 10^8$ | 33 (32/1) | 2200 |
| INALT10 | ORCA05 + $1404 \times 674$ | 2160/720 | $59 \times 10^8$ | 39 (37/2) | 4100 |
| INALT10x | ORCA05 + $1404 \times 924$ | 2160/720 | $78 \times 10^8$ | 44 (42/2) | 5300 |
| INALT20 | ORCA025 + $2799 \times 1839$ | 1200/400 | $560 \times 10^8$ | 104 (96/8) | 30500 |
| INALT20r | ORCA025 + $1804 \times 1024$ | 1200/400 | $230 \times 10^8$ | 104 (96/8) | 15000 |
| INALT60 | INALT20r + $2404 \times 1483$ | 900/300/100 | $1700 \times 10^8$ | 146 (134/12) | 121800 |

## 2.2 Numerical settings and parameterizations

All configurations share a range of settings described in the following. A "filtered" free surface formulation is used (Roullet and Madec, 2000) which damps fast external gravity waves. Assuming that sea surface height (SSH) anomalies are small compared to the resting depth, free surface is linearized which translates into a fixed ocean volume in time. The vertical mixing is parameterized according to a 1.5 turbulent kinetic energy closure (Blanke and Delecluse, 1993). Static instabilities are handled by enhancing diffusivities and viscosities by a factor of 10. Spatially varying Laplacian, iso-neutral mixing is applied to tracers, while momentum is subject to bilaplacian horizontal mixing. A diffusive bottom boundary layer formulation for tracers (Beckmann and Döscher, 1997), allowing a communication of two adjacent bottom grid cells at different depths levels, is active. Quadratic bottom friction is applied globally with an enhancement of a factor up to 50 to reduce the flow downstream of Torres Strait in ORCA05 and on the host grids of INALT10(x) and additionally downstream of Denmark Strait, and Bab el-Mandeb Strait in ORCA025 and on the host grids of INALT20(r) and INALT60. In the un-nested configurations as well as on the host grids of the nested configurations, the lateral boundary condition allows for free slip while no slip is allowed along the lateral boundaries of the oceans on the nest grids in all reference simulations. In all reference experiments, the wind stress formulation is taking into account the underlying ocean surface velocities ("relative winds"). For these two boundary conditions, sensitivity experiments are performed in INALT20 (see below). A second order centered tracer advection scheme (Total Variance Dissipation - TVD, Zalesak, 1979) is used. Momentum advection is in vector form with applied Hollingsworth correction (Hollingsworth et al., 1983). The vorticity term is formulated conserving both, the potential enstrophy of horizontally non-divergent flow and the horizontal kinetic energy (EEN, Arakawa and Hsu, 1990).

The nested configurations at 1/10° and 1/20° horizontal resolution share a spatial horizontal refinement factor of five with respect to their host grids, which would have implied an identical temporal refinement based on Courant-Friedrichs-Lewy

**Table 2.** Resolution dependent parameters for the different configurations: eddy induced velocity coefficient (aeiv0 [m$^2$s$^{-1}$]); horizontal eddy diffusivity coefficients for tracer (aht0 [m$^2$s$^{-1}$]) and momentum (ahm0 [m$^4$s$^{-1}$]) are nominally set for the maximum grid size (which is at the equator except for the nest grids in INALT20r and INALT60 that do not include the equator at high resolution) and scaled with the poleward decrease of the grids sizes; lateral mixing coefficient in the bottom boundary layer (ahtbbl [m$^2$s$^{-1}$]); sponge coefficients for tracer and dynamics (sponge [m$^2$s$^{-1}$]). Note that ORCA05 also represents the values for the host grid in INALT10 and INALT10x, ORCA025 those for the host grid in INALT20, INALT20r and INALT60. The settings for the first nest in INALT60, are identical to those given for INALT20(r).

|           | aeiv0 | aht0 | ahm0                   | ahtbbl | sponge |
|-----------|-------|------|------------------------|--------|--------|
| ORCA05    | 1000  | 600  | $-6.0 \times 10^{11}$  | 1000   | -      |
| ORCA025   | -     | 300  | $-1.5 \times 10^{11}$  | 1000   | -      |
| INALT10(x)| 0     | 120  | $-2.4 \times 10^{10}$  | 100    | 2700   |
| INALT20(r)| -     | 60   | $-6.0 \times 10^{9}$   | 40     | 600    |
| INALT60   | -     | 20   | $-6.7 \times 10^{8}$   | 4.444  | 200    |

(CFL) stability considerations. A temporal refinement of three has however proven to be stable. For the secondary nest in INALT60 both horizontal and temporal refinement factors are three against the first nest. A sponge layer is applied along the boundaries of the nests that damps (according to a 2nd order Laplacian operator) nest and host differences. It maintains consistency between the grid solutions, filters out noise that would develop along open boundaries. As explain by Debreu et al. (2008) this is a key ingredient in the overall nesting robustness. The host grids are updated with the three dimensional nest solution at every third host grid time step. The different model resolutions demand different parameters for the above mentioned schemes as summarized in Table 2.

A climatological river runoff is applied, distributing fresh water input from land along the coasts and at the estuaries of the 99 major rivers (Bourdallé-Badie and Treguier, 2006). At the rivers mouths and at the locations where runoff enters the ocean, enhanced vertical mixing over the upper 10 m is applied and the sea surface salinity restoring is suppressed. Elsewhere, extremely weak sea surface salinity restoring is applied via damping in the surface freshwater flux at a piston velocity of 50 m / 4 years (Griffies et al., 2009). Additionally, Newtonian tracer damping at the outflow of the Mediterranean Sea into the Atlantic is applied to correct the unrealistically shallow spreading of the Mediterranean Outflow. Damping is applied in a spherical area centered at 7° W, 36° N and 500 m depth with a horizontal radius of 1.5° and a vertical extent of 1000 m. Damping strength declines exponentially with increasing distance from the centre.

## 2.3 Hindcast and sensitivity experiments

The atmospheric forcing for all experiments described here is based on the COREv2 products and bulk formulae provided by Large and Yeager (2009). It builds on NCEP/NCAR reanalysis data merged with satellite-based radiation and precipitation, employing a set of parameter corrections to minimize global flux imbalances. 30-year long spin-up integrations, initialized

**Table 3.** To uniquely define the different model experiments, internal experiment identifiers are provided here.

| Configuration | Spin-up | Hindcast | Sensitivity experiments |
|---|---|---|---|
| ORCA05.L46 | KJH0003 | KJH0004 | |
| ORCA025.L46 | KJR36201h | KJR36216h | |
| INALT10.L46 | KJH0016 | KJH0017 | |
| INALT10x.L46 | KJH0006 | KJH0007 | |
| INALT20.L46 | KFS039 | KFS044 | KFS057 (FS_RW) KFS058 (FS_AW) KFS059 (FS_PW) |
| INALT60.L46 | KFS004 - KFS006 | | |

with termerature and salinity from the World Ocean Atlas (Levitus et al. (1998) with modifications in the polar regions from PHC, (Steele et al., 2001)) and an ocean at rest, forced by interannually varying atmospheric boundary conditions from 1980 to 2009, were performed in all configurations except for INALT60 where only ten years from 1980 to 1989 were integrated. The ocean states at the end of the spin-up integrations are used to initialize hindcast simulations for the period 1958 to 2009. All experiments are thus performed over the same integration length, utilizing the same atmospheric forcing. The comparisons between the different configurations therefore allow to isolate the impact of different resolutions. Despite COREv2 having a spatial resolution of $2° \times 2°$, being relatively coarse, it was chosen as the most coherent and robust dataset available at the time of the model integrations, to provide long (multi-decadal) hindcast simulations.

In INALT20, sensitivity experiments starting from the oceanic state at December 31st 1994 in the hindcast experiment and covering the period 1995 to 2009 are performed to test the influence of lateral and surface boundary conditions on the dynamics in the AC system, the South Atlantic and the overturning circulation. Building on the reference configuration with "no slip" on the nest grid (NS) and "free slip" on the host grid in all cases, sensitivity experiments are performed with "free slip" also in the nest (FS). Another set was performed to explore the role of ocean currents in the wind stress calculation. Compared to the original formulation as "relative winds" (RW), cases for "absolute winds" (not considering the ocean currents; AW) and "partial winds" (PW) are tested. In the latter, the influence of the ocean velocity in the wind stress formulation is reduced compared to the relative wind formulation by a factor of 0.7, consequently only taking into account 70% of the ocean currents (simplified after Renault et al. (2017)). While Renault et al. (2017) use a spatially varying coefficient for the influence of ocean currents in the wind stress formulation, a constant value, representative for the AC system is used here. Table 3 provides internal identifiers for the experiments used.

## 3 Results

The simulations described above are evaluated with respect to their general representation of mesoscale variability (Sect. 3.1). The impact of mesoscale processes onto the large-scale circulation is presented by the evaluation of the representation of key measures of the horizontal circulation in the South Atlantic, the western Indian and the Southern oceans (Sect. 3.2) as well

as of the meridional overturning circulation in the Atlantic Ocean (Sect. 3.3). The dependence on the resolution of mesoscale processes in the Agulhas Current system itself is analyzed in terms of simulating the Agulhas Current and Undercurrent (Sect. 3.4) as well as the Aguhas leakage (Sect. 3.5).

## 3.1 Mesoscale variability

The mesoscale is the dominating scale in the Agulhas region (Backeberg et al., 2008); its proper simulation is crucial for a resonable representation of the dynamics in a numerical model. As a measure for mesoscale activity, SSH variance from satellite observations is compared to the modelled solutions (Fig. 3). Satellite altimetry data are provided by AVISO interpolated to a 1/4° Mercator grid with daily resolution. The daily data have been averaged into 5-daily means prior to variance calculations over the 10-year long period 2000 to 2009 to allow for a direct comparison with the modelled SSH that is available at 5-daily resolution except for the experiment in ORCA025 where only monthly averages are available.

From observations (Fig. 3(f)), several highly active regions can be identified: The Mozambique Channel and the eddy path from the South East Madagascar Current towards the African coast, the meandering AC itself and, with highest activity, the retroflection area and the Agulhas Return Current. The path of Agulhas rings from the Indian into the Atlantic Ocean can be identified as elevated, although comparably weak, mesoscale activity. Aside from these regions, directly related to the AC, three further areas of enhanced activity stand out: the Andrew Bain Fracture Zone (ABFZ) (Ansorge and Lutjeharms, 2005) south of Africa at approximately 50° S / 30 ° E, the Malvinas Confluence Zone (MCZ) (Goni et al., 2011) on the other side of the Atlantic Ocean and a path of eddies entering the basin with the ACC.

In ORCA05, where the effect of eddies is parameterized, virtually no mesoscale activity is present (Fig. 3(a)). The eddy-permitting configuration ORCA025 (Fig. 3(b); based on monthly averages) shows elevated levels of SSH variance in the most prominent regions of mesoscale activity, the retroflection area and along the Agulhas Return Current as well as in the MCZ, but generally too weak. The AC and its upstream sources show no increased sign of mesoscale variability. However, in agreement with findings by Quartly et al. (2013), an accompanying experiment (not shown) with no slip along the lateral boundaries exhibits elevated mesoscale variability here, due to the evolution of eddies in and propagation out of the Mozambique Channel. With increasing resolution, the representation of these large scale features improves in strength and structure. INALT10(x) (Fig. 3(c), (e)) and INALT20 (Fig. 3(d)) explicitly simulate variability in the source regions, and in consequence a meandering and more variable AC.

The inflow of ACC eddies is adumbrated in ORCA025 but evolves to a distinct path only in INALT20 (Fig. 3(d)) where additionally, the mesoscale activity in the MCZ becomes stronger and improved with respect to its spatial structure. The ABFZ as well as the Mozambique Channel and the area southwest of Madagascar, begin to emerge as highly active regions, although still weaker compared to observations. INALT10x (Fig. 3(c)) also resolves mesoscale features, although weaker than in INALT20, in the specified areas, except for the ACC inflow, that obviously evolves in a region located outside the nest in the non-eddying host grid. INALT10 (Fig. 3(e)), with its nested area limited to north of 50° S covers the variability around southern Africa, hence compares to INALT10x but lacks all mesoscale activity south of it. In consequence, it shows a different and weaker activity in the MCZ. All eddying simulations show a northwestward propagation of Agulhas rings into the Atlantic

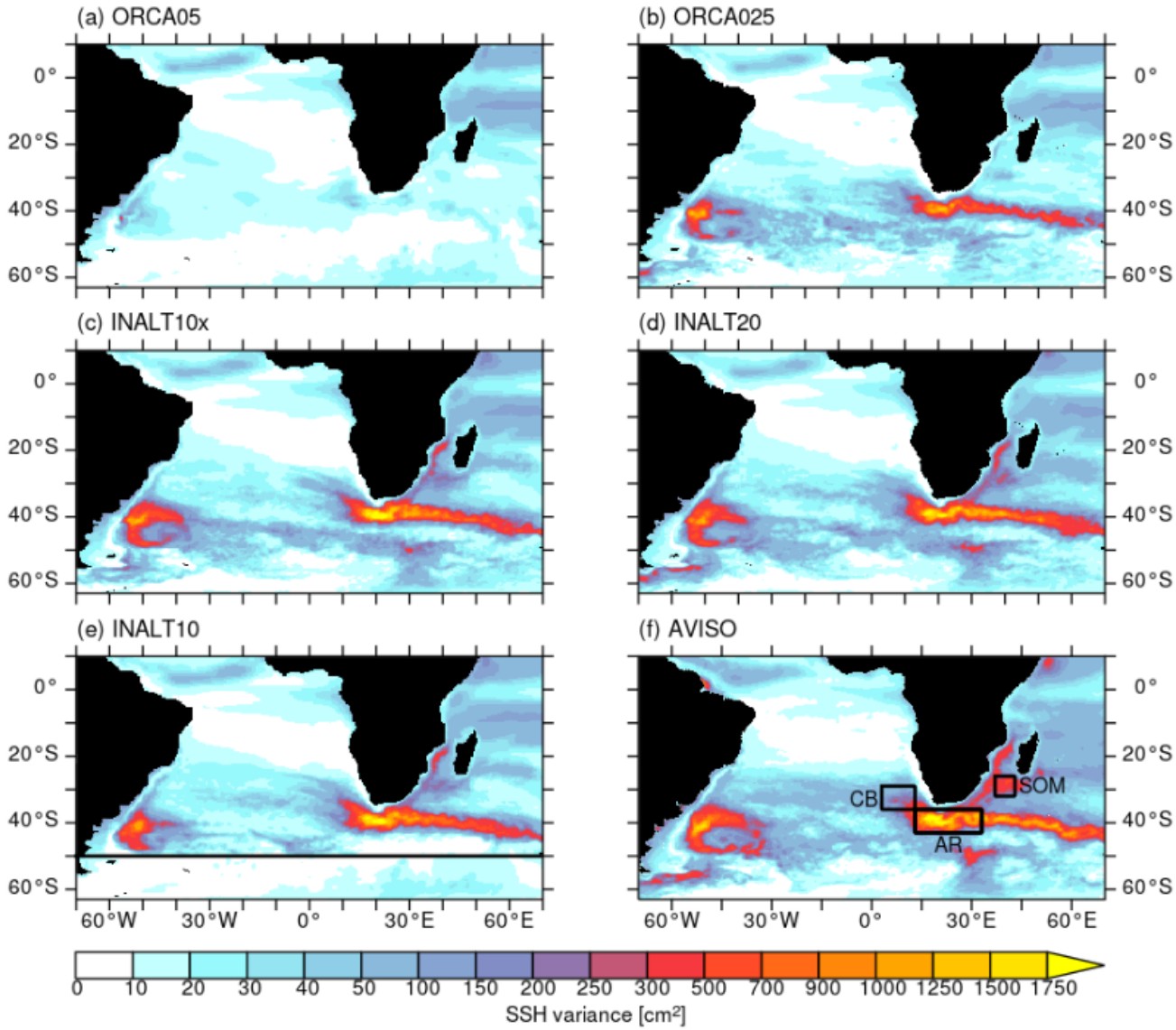

**Figure 3.** SSH variance [in $cm^2$] for the period 2000-2009 based on 5-day averages from (f) satellite altimetry data (AVISO) compared to the hindcast experiments in the different configurations (a) ORCA05, (b) ORCA025 (based on monthly averages), (c) INALT10x, (d) INALT20 and (e) INALT10. The horizontal line in (e) at 50° S indicates the southern boundary of the nested region in INALT10; the boxes in (f) mark the areas used for time series analyses (see Fig. 5). Note: the discrepancy in the temporal resolution for ORCA025 does not alter the findings (not shown).

Ocean, while in the satellite observations, a rather zonal band between 30° S and 40° S of elevated SSH variance marks their path.

In the greater AC system, mesoscale activity is comparable in INALT20 and INALT60 (Fig. 4(a and b)), with INALT20 showing a stronger variability in the Mozambique Channel and weaker levels in the retroflection area. INALT60 shows a slightly wider ring path into the Atlantic Ocean, although this can also be due to the short timespan (three years) used for the analysis.

Not only the resolution, but also different parameterizations have an impact on the representation of the mesoscale variability as exhibited by sensitivity experiments in INALT20. Both, the lateral boundary condition and the wind stress formulation have been modified. While changing the lateral boundary condition from no slip (Fig. 4(c)) to free slip (Fig. 4(d)) only slightly increases SSH variance southwest of Madagascar and in the ABFZ and reshapes the patch in the MCZ, changing the wind formulation from relative to absolute winds (Fig. 4(e)) has a significant impact on the distribution of mesoscale variability. Although SSH variance is better represented in the Mozambique Channel and southwest of Madagascar as well as in the ABFZ and the MCZ in the simulation under absolute winds, the observed patterns south of Africa and along the Agulhas Return Current are overestimated in strength and extent resulting in unrealistically high variability. This also leads to a too regular and too pronounced path of Agulhas rings being shed into the Atlantic Ocean, a well known behaviour (e.g. Biastoch et al., 2008c; Barnier et al., 2006) that is unacceptable for simulations being used to analyze AC and leakage influences onto the Atlantic Ocean. An attempt to overcome the deficits of the two simulations under relative and absolute winds, respectively, is given by the partial wind experiment. It shows no significant improvement compared to the relative wind experiment (Fig. 4(f)), despite a slight elevation of variability in and south of the Mozambique Channel and in the MCZ. In consequence, we continue with the no slip lateral boundary condition on the nest grid and relative winds for the reference experiment.

Timeseries of SSH variance, computed as the variance of 5-daily data within a 5 year long running window provide a measure for the temporal variability in mesoscale activity. It is depicted for observations and the hindcast simulations in the nested configurations in Fig. 5 for three areas around South Africa: southwest of Madagascar (SOM), the Agulhas retroflection area (AR) south of Africa and the Cape Basin (CB) west of South Africa (boxes marked in Fig. 3(f)). It exhibits different types of variability. SSH variance southwest of Madagascar (Fig. 5(a)), dominated by variations originating in the Mozambique Channel, is marked by interannual variability with a period of approximately seven years in all nested solutions. In this region, the SSH variance in INALT20 is on a slightly higher mean level than in the simulations at 1/10° horizontal resolution but still strongly underestimates the observations (Table 4). The corresponding timeseries of SSH variance anomalies (Fig. 5) exhibit that the variability in the early period of the simulations is comparable among the different configurations, while in the subsequent decades, INALT10x stands out with reduced variations leading into a very strong increase towards the end of the simulation, in agreement with the observed anomalies, even exceeding the solution of INALT20. In this region, only INALT20 is significantly correlated with observations (r=0.95; see Table 5) in the overlapping period 1995 to 2005 while the modelled solutions are all significantly correlated among themselves for the short period, mainly due to the decrease in the late 1990 followed by an increase in the 2000s.

In the Agulhas Retroflection Area (Fig. 5 (b)), influenced by the AC and its meandering, the dominant period of variability is on decadal time scales in INALT10 and INALT10x and even longer in INALT20. The absolute values are comparable to, but

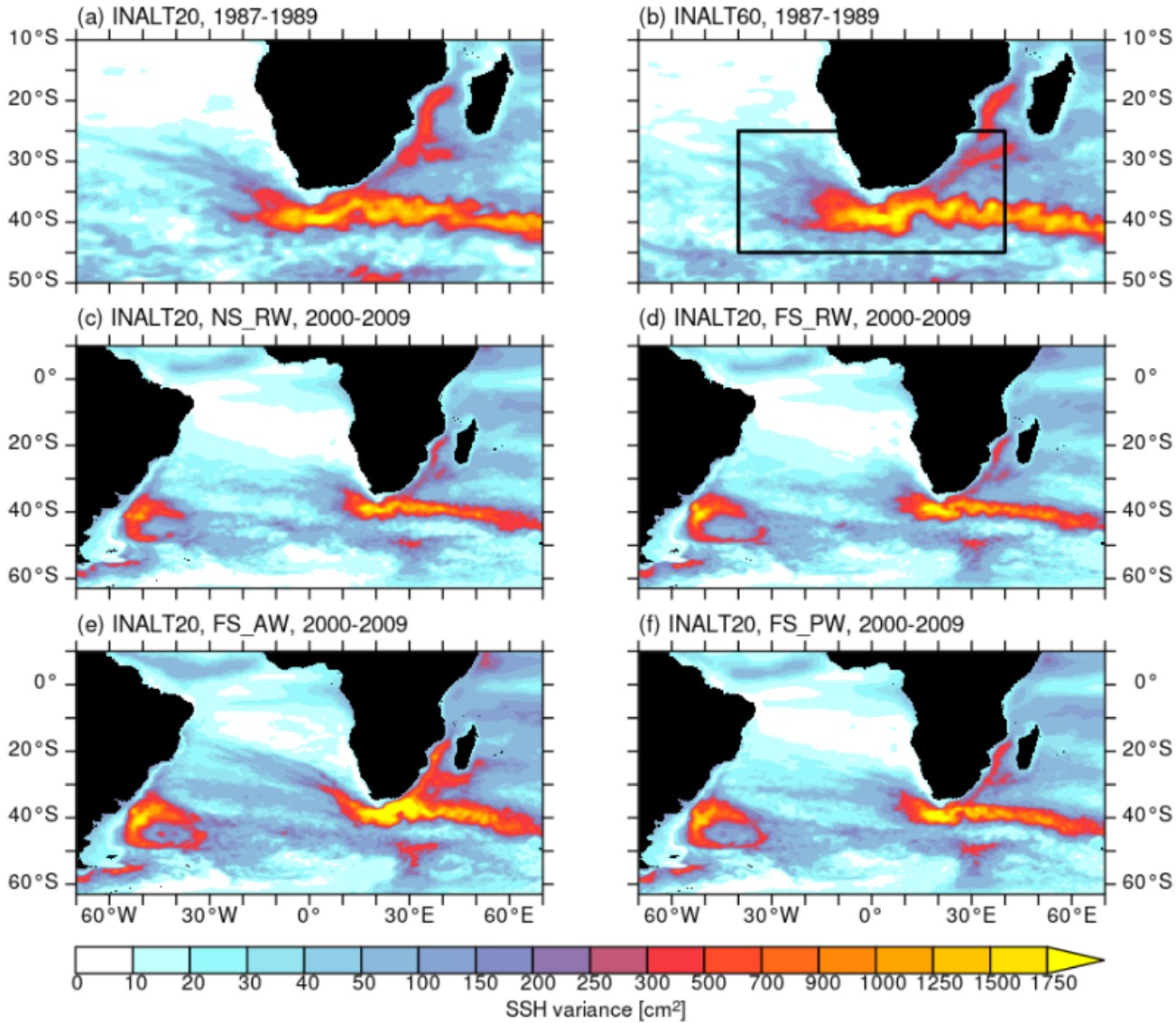

**Figure 4.** SSH variance [in $cm^2$] for the period 1987 to 1989 from the spin-up experiments in (a) INALT20 and (b) INALT60 based on 5-day averages; The black box in (b) marks the region of the secondary 1/60° nest in INALT60. (c to f) give the SSH variance [in $cm^2$] for the period 2000-2009 based on 5-day averages from the different sensitivity experiments in INALT20: (c) no slip, relative winds, (d) free slip, relative winds, (e) free slip, absolute winds, (f) free slip, partial winds.

still underestimating the observations by 15-20%. Here, only INALT10 is significantly correlated with observations (r=0.90, see Table 5). As southwest of Madagascar, all model simulations are significantly correlated in the observational period.

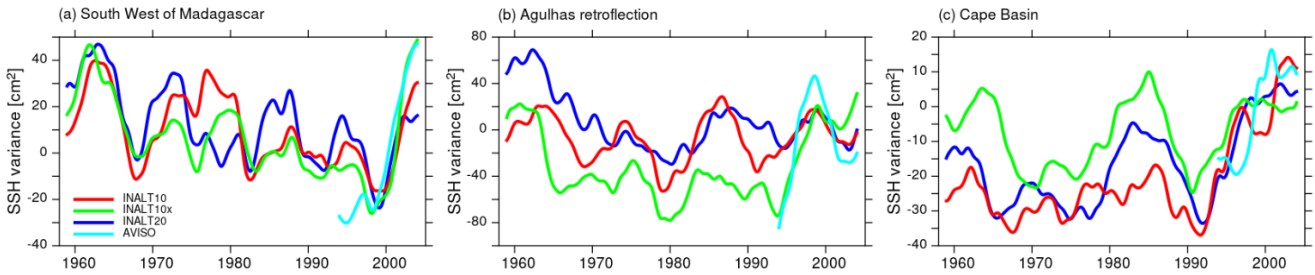

**Figure 5.** Timeseries of SSH variance anomalies to the 1995-2005 mean [in $cm^2$] computed in a 5 years running window based on 5-daily averages for the three different regions indicated in Fig. 3(f)) from the nested configurations INALT20 (blue) INALT10 (red), INALT10x (green) and from AVISO-satellite altimenty (light blue). The curves are interannually filtered. For mean values and correlations see Tables 4 and 5, respectively.

In the Cape Basin (Fig. 5 (c)) the dominant variability is on decadal time scales in all simulations at comparable levels. In this region for the observational period, the variability on interannual timescales is un-correlated, both between observations and the modelled solutions as well as between the different configurations. However, on longer time scales the correlations among the modelled solutions increase with INALT10x and INALT20 being significantly correlated at 95% confidence level (not shown). In contrast to the regions south of Madagascar and in the Agulhas retroflection area, the Cape Basin is located outside the most eddy-active region of the AC system within the path of the Agulhas rings, shedded into the Atlantic Ocean.

All configurations have a similar pattern of SSH variance for the three regions and all underestimate the mean, with INALT20 being closest to observations. The remarkable co-variability on decadal time scales most probable stems from the common surface forcing. A detailed understanding of this is part of an individual research topic in itself and will be reconsidered elsewhere. In the greater Agulhas region a horizontal resolution of 1/10° locally appears to be enough to simulate mesoscale variability at a reasonable level whereas the lack of mesoscale activity outside the nested region in INALT10 and INALT10x leads to an underestimation of mesoscale variability entering the South Atlantic through Drake Passage and therefore in the MCZ and ABFZ. Among the configurations presented here, INALT20 simulates all the prominent mesoscale features within the region of interest most reasonably.

**Table 4.** Mean (2000-2009) SSH variance [in $cm^2$] in the three different regions indicated in Fig. 3(f): southwest of Madagascar (SOM) | Agulhas Retroflection (AR) | Cape Basin (CB). See Fig. 5 for corresponding anomaly timeseries.

|  | SOM | AR | CB |
|---|---|---|---|
| Observations | 331 | 794 | 160 |
| INALT10 | 150 | 637 | 127 |
| INALT10x | 160 | 662 | 107 |
| INALT20 NS_RW | 183 | 654 | 127 |
| INALT20 FS_RW | 180 | 765 | 131 |
| INALT20 FS_AW | 374 | 1197 | 248 |
| INALT20 FS_PW | 207 | 835 | 176 |

**Table 5.** Correlations in SSH variance timeseries (yearly) (see Fig. 5) between AVISO and the nested configurations as well as among the nested configurations for the period 1995-2005 for the three regions indicated in Fig. 3(f): Southwest of Madagascar (SOM) | Agulhas Retroflection (AR) | Cape Basin (CB). Significant correlations (at 95% confidence level) are given in bold.

|  | INALT10 | INALT10x | INALT20 |
|---|---|---|---|
| AVISO | 0.86 \| **0.90** \| -0.73 | 0.84 \| 0.87 \| -0.24 | **0.95** \| 0.87 \| 0.32 |
| INALT10 |  | **0.99** \| **0.97** \| -0.30 | **0.88** \| **0.93** \| -0.21 |
| INALT10x |  |  | **0.88** \| **0.93** \| 0.24 |

### 3.1.1 Towards resolving the sub-mesoscale - an outlook

INALT60 with horizontal grid spacing below 2 km provides the necessary horizontal resolution to resolve scales down to 10 km. In addition to the horizontal resolution, the vertical grid, the spatial and temporal resolution of the atmospheric forcing and the diffusion setup strongly control the simulation of the smaller scale flows in the model. Developments to improve the configuration in the future are ongoing and are presented by Schubert et al. (under review). Here, a short outlook is given.

To adequately simulate the ocean currents potentially resolvable with the given horizontal grid, the vertical resolution needs at least to resolve the vertical structure of the corresponding horizontal flow which can be approximated by the baroclinic modes (Stewart et al., 2017). Based on hydrographic measurements, Stewart et al. (2017) provide a reference for the required vertical resolution to resolve the first, second and third baroclinic modes. A new vertical grid, that shall resolve the third baroclinic mode, with 120 vertical levels and a grid spacing of 1 m near the surface, 10 m at about 350 m depth and 100 m in the deep ocean is under development. A very high-vertical resolution is required in the mixed layer, as sub-mesoscale currents mainly occur in boundary layers and are surface intensified (McWilliams, 2016). The mixed layer depth in the mid-latitudes regions of the storm tracks, such as the Agulhas region, is associated with a strong seasonal cycle. Weaker wind-stress in summer leads to a thin mixed-layer of down to less than 20 m, while stronger wind-stress in winter leads to a thick mixed-layers exceeding 150 m (de Boyer Montégut et al., 2004). To resolve smaller scale flows within the mixed-layer also during summer time, at least 10 vertical levels in the uppermost 20 m are required.

Near-surface sub-mesoscale features, resolvable with a 1/60° configuration, evolve at time scales less than a day (McWilliams, 2016). Due to the sub-daily timescales, a realistic representation of the daily cycle in the forcing fields is necessary for simulating the sub-mesoscales properly. The CORE atmospheric forcing is given at relatively coarse resolution in space ($2°\times2°$) and time (6 hourly for the highest resolved fields). For simulations with INALT60, the higher resolved JRA55-do forcing (Tsujino et al., 2018), which is associated with relatively higher horizontal ($0.5°\times0.5°$) and temporal (3 hourly) resolutions, will be used in the future.

Both the numerical diffusion associated with advection schemes for tracer and momentum and the explicit diffusion in the primitive equations contribute to the model diffusion. So far, the same model diffusion setup is used for INALT60 and INALT20: TVD scheme and Laplacian explicit diffusion with constant diffusion coefficient for tracers and Vector Invariant advection scheme with EEN vorticity formulation supported by bi-Laplacian explicit viscosity with a constant viscosity coefficient for momentum. The diffusion coefficient has been linearly (quadratically) scaled down with the grid spacing for tracers (momentum). The coefficient has thereby to be large enough to prevent the strongest simulated shears from numerical instability. A disadvantage of this method is that moderate shears that would not have lead to numerical instability are also damped. In the future, third-order upstream biased schemes (UBS, Webb et al., 1998; Farrow and Stevens, 1995; Madec and the NEMO team, 2014) for tracer and momentum will be used. These schemes are numerically diffusive enough to inhibit numerical instabilities at the grid-scale. Explicit diffusion with UBS is only needed, if the numerical diffusion is not large enough to be realistic. The evaluation of the model performance with UBS and a respective validation against observations is presented by Schubert et al. (under review).

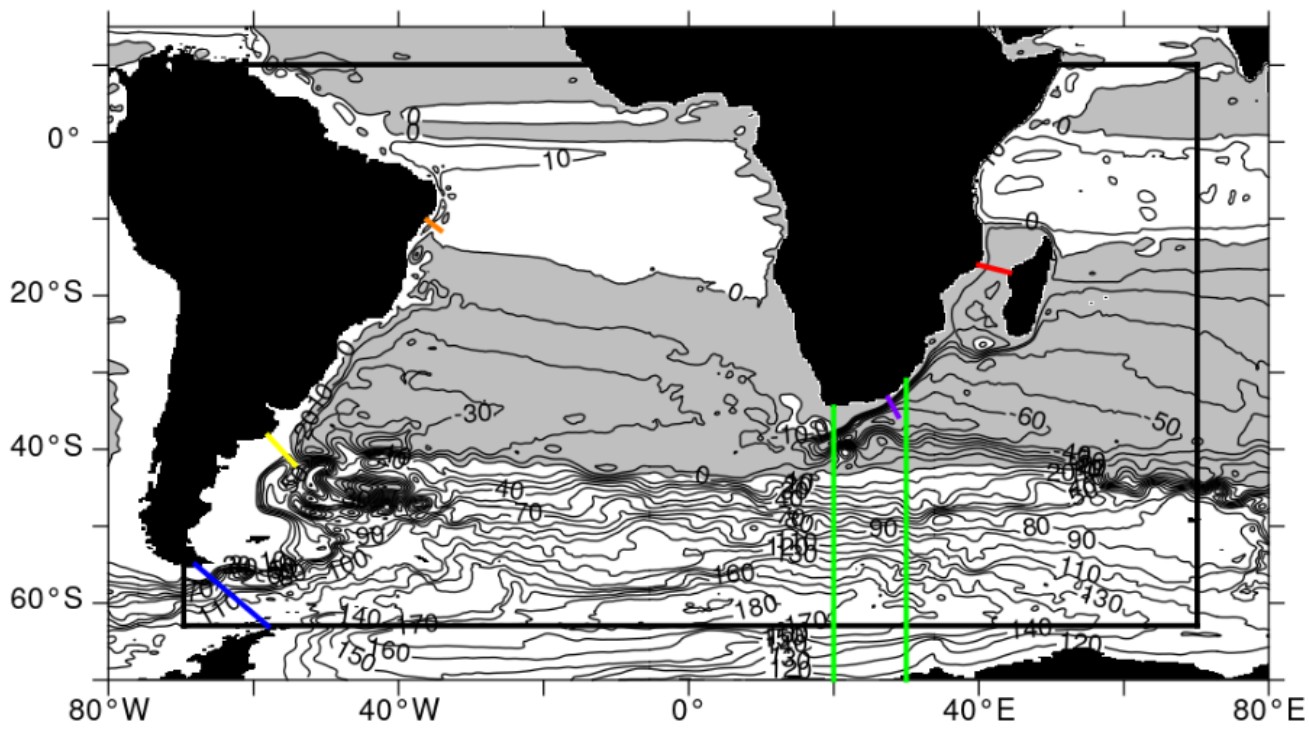

**Figure 6.** Mean (2000-2009) barotropic stream function as simulated within INALT20 (nested region demarcated by the black box). Grey shading indicates anti-cyclonic circulation; contours are in 10 Sv intervals. The band for ACC transport calculations south af Africa is marked by the green lines; Drake Passage is indicated in blue; the location of the LOCO array is given in red, the section crossing the Malvinas Current in yellow and the section crossing the North Brazil Current in orange. The purple line indicate the location of the ACT array that is referred to in Sect. 3.4.

## 3.2 Horizontal gyre circulation and transports

An important consequence of the open setting of the AC system is the connection between the subtropical gyres in the Indian and Atlantic oceans. This connection south of Africa represented by the Agulhas leakage (see Sect. 3.5) is part of the Southern Hemisphere "supergyre" (Speich et al., 2007). Only a certain portion of the water arriving in the South Atlantic through the AC system becomes part of the AMOC. According to models, about 40-50% recirculate in the horizontal gyre circulation in the South Atlantic at least once (Rühs et al., 2013, submitted).

All model configurations share the same structure and extent of the supergyre (Fig. 6, only INALT20 is shown exemplarily), which is, to first order, a consequence of the surface forcing fields. The horizontal streamfunction conveniently provides measures of transports across individual sections. Below we evaluate transports of the ACC, and through the Mozambique Channel.

Furthermore, the Malvinas Current (MC) and the North Brazil Current (NBC), two important western boundary currents in the South Atlantic, provide a measure for transport in the subtropical gyre in the South Atlantic and the interhemispheric transport from the south towards the North Atlantic (Rühs et al., 2015). The MC has been suggested to play a significant role for the upper limb of the Atlantic Meridional Overturning (Fricourt et al., 2005): North of the presented section, the MC collides with the southward Brazil Current forming the Brazil-Malvinas Confluence Zone, one of the regions in the world's oceans with highest eddy activity (see analyses of SSH variance in Sect. 3.1). The current is suggested to then transport water across the Atlantic Ocean towards the east, entering the South Atlantic Current and, further downstream, the NBC which transports the water across the equator (Hummels et al., 2015).

### 3.2.1   Transport of the ACC

The mean ACC transport south of Africa, estimated by the maximum in the barotropic stream function between 20° E and 30° E, following Durgadoo et al. (2013), is 175 Sv and 178 Sv in the non-eddying configurations ORCA05 and INALT10 respectively, and 20 Sv stronger if eddies are (at least partially) resolved in ORCA025, INALT10X and INALT20 (Fig. 7(a)). Evans et al. (2017) review a series of observational estimates of the ACC transport across 30° E resulting in a range between 131.7 Sv and 160 Sv. However, the difference in the transport from the simulations here is mainly due to a recirculation gyre in the Weddell Sea, rather than to the different strength of the ACC (Fig. 6). The transport through Drake Passage shows a different picture (Fig. 7(b)). In INALT10x and INALT20 Drake Passage is located directly at the margin of the nested area whereas in INALT10, whose southern boundary is located at 50° S, it is outside of the nest. The simulated mean transport is weaker (∼117 Sv) within the configurations with an eddying representation of the Drake Passage (ORCA025 and INALT20) than with a non-eddying representation (ORCA05 and INALT10; ∼122 Sv). The latter two configurations however separate over time, with INALT10 being subject to a stronger decline of the ACC. In contrast to INALT10, INALT10x shows a higher transport (∼133 Sv), due to a larger geostropic component across the ACC. Given the current uncertainty in observational values for the Drake Passage transport ranging from 124.7±9.9 Sv (Whitworth and Peterson, 1985) via 136.7±7.8 Sv (Cunningham et al., 2003) to 173.3±10.7 Sv (Donohue et al., 2016), it is difficult to identify the more realistic configurations, however the estimates from all configurations are at the lower end. On interannual timescales, the simulated transport time series are significantly correlated among all the configurations with highest values > 0.97 between ORCA05 and INALT10(x), 0.91 between ORCA025 and INALT20 and 0.89 between INALT10x and INALT20. The reduction in correlation values is due to non-linearities, decoupling the individual transports from the common wind forcing.

### 3.2.2   Transport through the Mozambique Channel

All simulations exhibit the same interannual variability and declining trend of the barotropic transport through the Mozambique Channel but with different long term mean states. Although all runs experience a similar long-term decline of 1.5 Sv per decade, it is to note that an accompanying ORCA025 experiment with repeated-year forcing is not subject to a significant trend (not shown). Consequently, this trend is rather not an effect of model drift but likely to be related to trends in wind stress and wind-stress curl over the Indian Ocean (DiMarco et al., 2002).

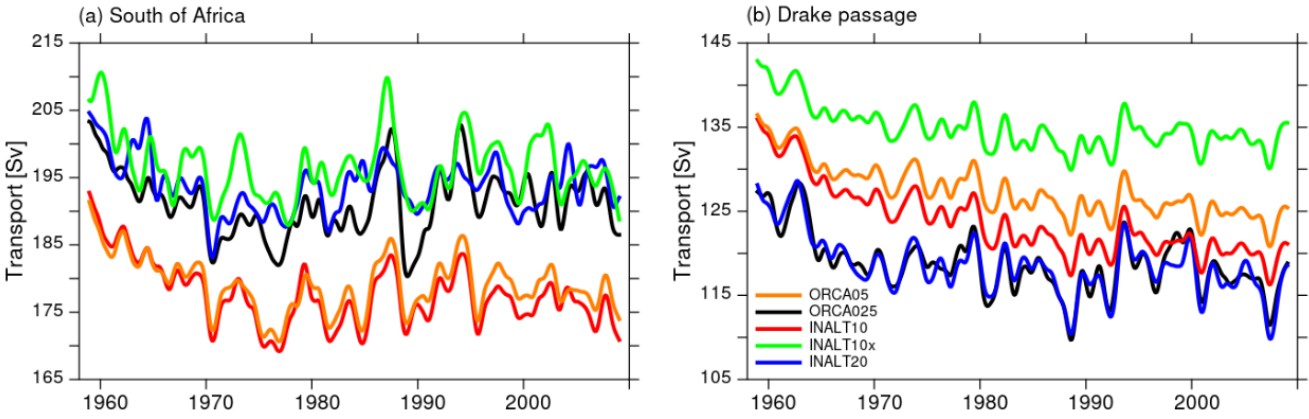

**Figure 7.** Timeseries of ACC transports [in Sv] as given by the maximum in the barotropic stream function south of Africa between 20° E and 30° E (a) and the barotropic transport through Drake Passage (b) for ORCA025 (black), INALT20 (blue), ORCA05 (orange), INALT10 (red) and INALT10x (green)

At 17° S a mooring array of the "Long-term Ocean Climate Observations (LOCO) Indian Ocean" program has been established since 2003 (Ridderinkhof et al., 2010), providing an observational time series of the net transport through the Mozambique Channel. The comparison between these observed and the simulated transport anomaly timeseries of the different configurations exhibits a strong correspondence on seasonal to interannual timescales (Fig. 8(b)). Before mid-2006, when a transition from a phase that is marked by high variability, to a phase with less fluctuations in the Mozambique Channel transport (van der Werf et al., 2010; Ullgren et al., 2012), the higher-frequency fluctuations are underestimated by all model configurations. After this transition in 2006, all simulations capture the variability quite well, except for some observed extrema like in July 2008 or June 2009. The mean values, however, are underestimated in the high-resolution configurations with only 10.0 Sv (INALT10) and 11.6 Sv (INALT10x and INALT20) compared to the observed 16.4 Sv (Table 6). The un-nested models do better in representing this mean transport with 16.1 Sv in ORCA025 and an overestimating 17.4 Sv in ORCA05. While in the non-eddying configurations this is an effect of the free slip solution and the unrealistic current in the Mozambique Channel, the underestimation in the eddying configurations points to a still underrepresented transport in the Mozambique eddies (van der Werf et al., 2010).

In the sensitivity experiments with INALT20, free slip lateral boundary conditions cause an increase in mean transport by about 50% (Table 6), while the variability remains comparable.

The seasonal cycle of the Mozambique Channel transport (Fig. 8(c)) is very robust among the different simulations and is in agreement with observations. van der Werf et al. (2010) ascribe the seasonal cycle to yearly variations in the wind forcing west of 75° E. It is interesting to note that a major part of the good correlation between the simulated and observed values is due to the realistic representation of this seasonal cycle (Table 6).

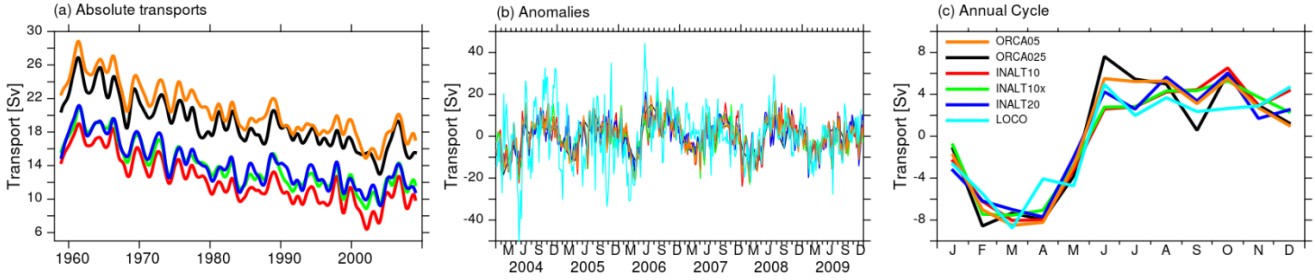

**Figure 8.** Timeseries of the southward barotropic transport through the Mozambique Channel [in Sv] as absolute values (a) and anomalies (b) compared to observations. The seasonal cycle of the anomalies for the period 2004 to 2009 is given in (c).

**Table 6.** Mean (2004-2009) transports through the Mozambique Channel and correlations between observed and modelled transport time-series based on monthly averages including|subtracting the seasonal cycle. Correlations, significant at 95% confidence level are given in bold.

|                | MOZ  | Correlations      |
|----------------|------|-------------------|
| Observations   | 16.5 |                   |
| ORCA05         | 17.4 | **0.62** \| **0.53** |
| ORCA025        | 16.1 | **0.53** \| 0.36  |
| INALT10        | 10.0 | **0.61** \| **0.47** |
| INALT10x       | 11.6 | **0.62** \| **0.50** |
| INALT20 NS_RW  | 11.6 | **0.62** \| **0.51** |
| INALT20 FS_RW  | 14.4 | **0.49** \| 0.23  |
| INALT20 FS_AW  | 15.5 | **0.61** \| 0.45  |
| INALT20 FS_PW  | 15.3 | **0.63** \| **0.52** |

### 3.2.3 Malvinas Current

The MC originates in the Drake Passage as the northern branch of the ACC, turning northward along the South American continental slope into the Atlantic Ocean. It consists of multiple jets (Piola et al., 2013) that align towards the north into a coherent current at 41° S whose mean structure and temporal variability have been assessed based on mooring data and satellite altimetry (Artana et al., 2018). Here, a comparison with the mean structure of the Malvinas Current as simulated in the different configurations is given (Fig. 9). ORCA05 and INALT10 both lack a MC at 41° S, due to a recirculation towards the East loacted further south that can be attributed to the missing resolution in the ACC and a consequently weaker MC. The configurations covering the entrance of the ACC into the Atlantic basin at eddying resolution (ORCA025, INALT10x and INALT20) all show a wide band of northward surface intensified velocities with a core located between 70 km and 80 km

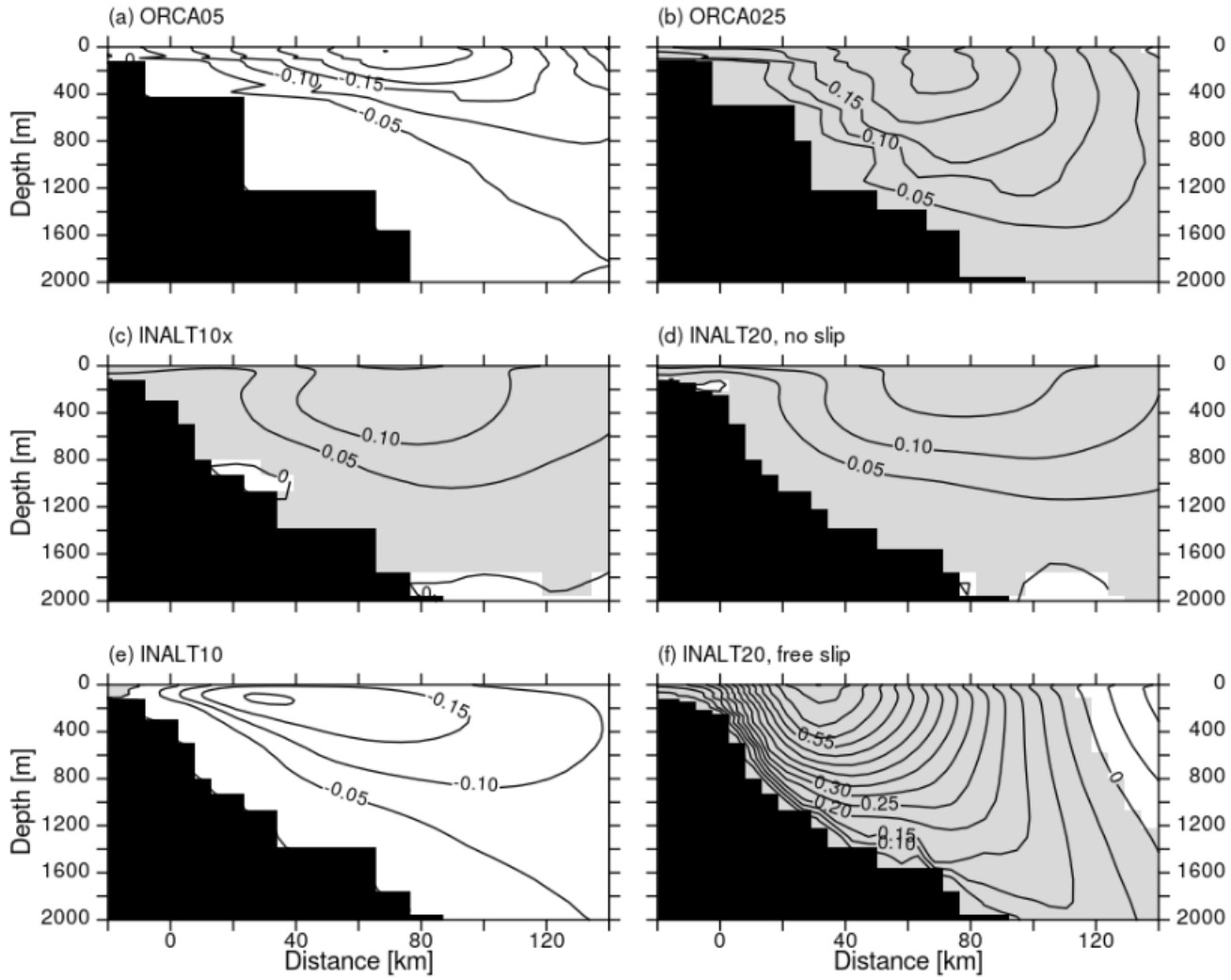

**Figure 9.** Mean (2000-2009) MC structure at 41° S from ORCA05 (a), ORCA025 (b), INALT10x (c), INALT20 NS_RW (d), INALT10 (e) and INALT20 FS_RW (f). Note that southwestward/negative velocities are shaded in white, northeastward/positive in grey, contoured are intervals of 5 cm s$^{-1}$

off-coast. The mean transports increase with increasing resolution from 25 Sv in ORCA025 to 37 Sv in INALT20 (see Table 7 for all mean transports). The latter well fits the observed transport, although featuring weaker velocities over a wider current core than in observations (Artana et al., 2018, their Fig. 5 (a)).

The sensitivity experiments in INALT20 show, due to the free slip condition, a current core, that is located closer to the coast, with a structure that better resembles the observed (Fig. 9(f)) but with overestimated velocities leading to mean transports of roughly 48 Sv independent from the applied wind stress formulation (Table 7).

**Table 7.** Mean (2000-2009) transports [m s$^{-1}$] of the Malvinas Current (MC) at 41° S and the North Brazil Current (NBC) at 11° S. Observational values are from Artana et al. (2018) and Hummels et al. (2015), respectively.

|  | MC | NBC |
|---|---|---|
| Observations | 37.1 ± 2.6 | 23 ± 3 |
| ORCA05 | 10.3 | 13.7 |
| ORCA025 | 25.1 | 17.7 |
| INALT10 | 16.7 | 19.3 |
| INALT10x | 32.5 | 18.8 |
| INALT20 NS_RW | 37.2 | 20.5 |
| INALT20 FS_RW | 48.5 | 18.8 |
| INALT20 FS_AW | 47.8 | 19.4 |
| INALT20 FS_PW | 47.4 | 18.3 |

### 3.2.4 North Brazil Current

The NBC forms a bottleneck for water from the South Atlantic on its way into the North Atlantic and is therefore crucial for the inter-hemispheric exchange and the AMOC (Rühs et al., 2015; Hummels et al., 2015). The representation of the NBC in the different model configurations is evaluated by comparing the current structure and mean transports with observations at 11° S (Fig. 10). The observed NBC shows a subsurface core, wherefore it is also referred to as North Brazil Undercurrent, located at 35.7° W off the coast of Brazil. The current reaches off-coast to approximately 35.2° W and down to 1200 m with core velocities of roughly 0.5 m s$^{-1}$ (Hummels et al., 2015, their Fig. 1b). The mean NBC transport is evaluated to 23±3 Sv. The NBC in ORCA05 is located further off-shore with underestimated velocities leading to a mean transport of only 13.7 Sv. With increasing resolution, the mean transport also increases, up to 20.5 Sv in INALT20 (see Table 7 for all transport values). The current structure also improves with increasing resolution, with the current core moving towards the coast and the observed tilted shape becoming more pronounced.

The NBC in the sensitivity experiments in INALT20 is of slightly weaker strength when compared to the reference experiment (Table 7) with a more coastally confined current structure due to the free slip condition (Fig. 10(f)). As for the MC, the wind stress formulation is of minor importance for the representation of the NBC.

The strengths of the western boundary currents in the South Atlantic are represented reasonably well in INALT20, while the configurations that have a coarser resolution at these locations underestimate the mean current transports. The sensitivity experiments in INALT20 exhibit a strong dependency of the MC structure and strength on the lateral boundary condition while the NBC is comparably robust.

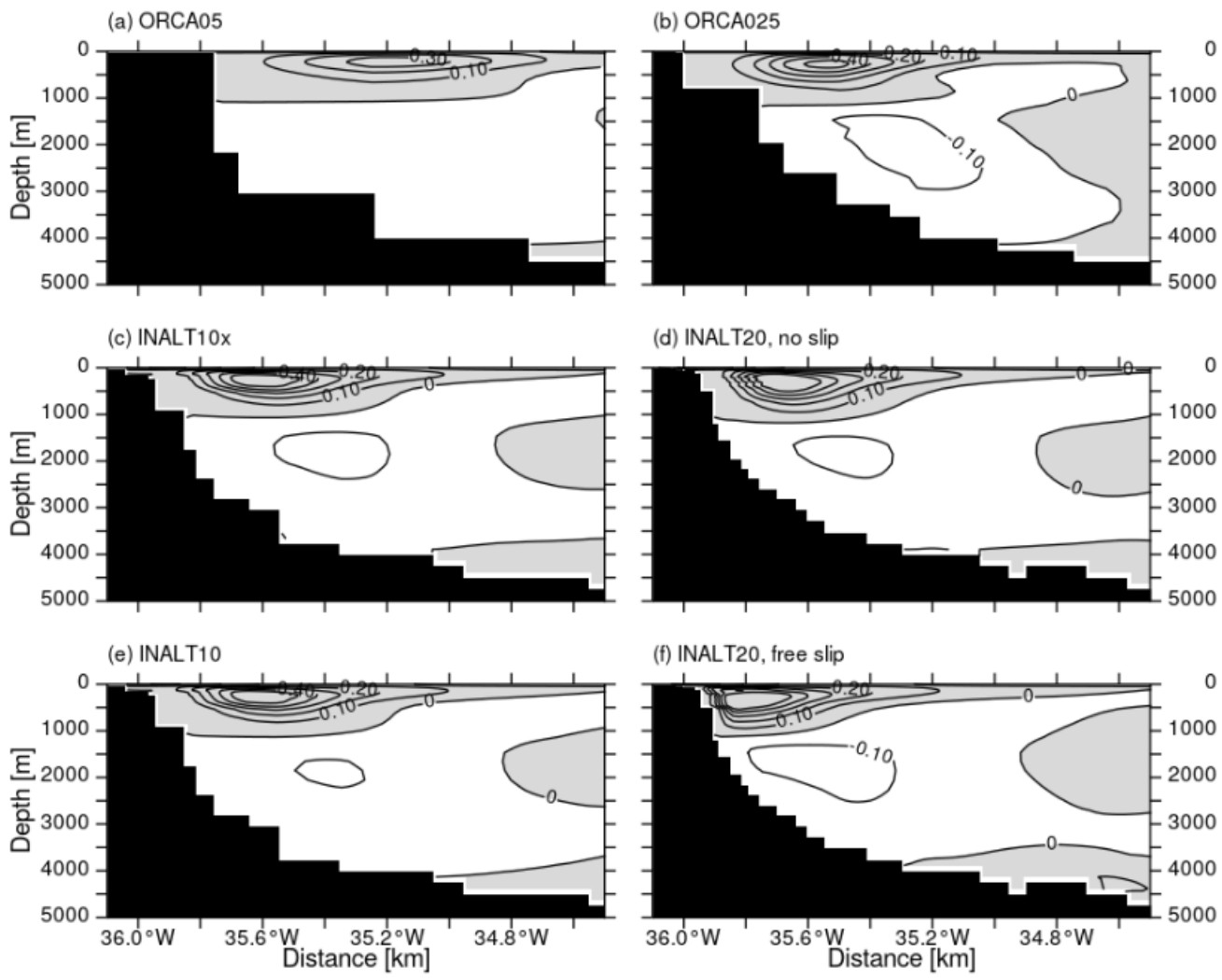

**Figure 10.** Mean (2000-2009) NBC structure at 11° S from ORCA05 (a), ORCA025 (b), INALT10x (c), INALT20 NS_RW (d), INALT10 (e) and INALT20 FS_RW (f). Note that southwestward/negative velocities are shaded in white, northeastward/positive in grey, contoured are intervals of 10 cm s$^{-1}$

## 3.3 Atlantic meridional overturning circulation

Owing to the interoceanic exchange south of Africa, the AC system plays a role in the global overturning circulation (Gordon, 1986). Here, surface and intermediate waters are transported from the Indian Ocean into the Atlantic (Biastoch and Böning, 2013), and eventually become part of the AMOC (Rühs et al., 2013; Biastoch et al., 2009b). On decadal timescales, AMOC anomalies with origin in the AC system are reflected in the AMOC in the South and North Atlantic (Biastoch et al., 2008a). On multi-decadal timescales, Agulhas leakage impacts the Atlantic hydrography and heat content (Biastoch et al., 2015; Lübbecke et al., 2015). A proper representation of the AMOC is therefore crucial to study not only the embedment of the AC system within the global circulation but also the feedback on the North Atlantic.

The simulated mean structure of the zonally integrated meridional velocities is similar among all simulations (Fig. 11), representing the expected structure: an upper cell with northward transport above $\sim$1000 m depth, sinking north of $\sim$40°N, and southward flow below down to $\sim$3500 m depth, transporting North Atlantic Deep Water - the North Atlantic Deep Water cell. Underneath, Antarctic Bottom Water (AABW) invades the Atlantic Basin from the south, forming the deep AABW cell with northward transport below $\sim$4000 m depth. Both cells increase in strength with increasing horizontal resolution of the model configurations.

With increased horizontal grid resolution either on the global scale, from ORCA05 to ORCA025, but also from the host to the respective nests, from ORCA05 to INALT10x and from ORCA025 to INALT20, the strength of the upper and lower overturning cells increases. For INALT10 this relationship holds only for the upper overturning cell while the AABW cell is slightly weaker represented than within ORCA05. Due to the reduced southward extend of the high-resolution nest in INALT10 this can be attributed to the missing effect of resolved transport of AABW across the subtropical front. The same argument accounts for the strengthening of the lower overturning cell comparing INALT10x and INALT20, implying, the need of resolutions beyond 1/10° to represent the impact of the mesoscale on deep water formation in the Southern Ocean. Changing the lateral boundary condition in INALT20 from no slip to free slip shows a strenghtening of the deep AABW cell. Consequently the stronger AABW cell slightly lifts, and more important, dampens the upper overturning cell as suggested by Frajka-Williams et al. (2011) from hydrographic observations at 24.5° N as well as from modelling studies (e. g. Swingedouw et al. (2009); Martin et al. (2015)).

The model simulations all underestimate the observed AMOC transport. In the South Atlantic, located at 34.5°S (Ansorge et al., 2014), the South Atlantic MOC Basin-wide Array (SAMBA, embedded in the South Atlantic MOC (SAMOC) initiative (Garzoli et al., 2013)) started to continuously monitor the meridional overturning circulation in 2002. Meinen et al. (2018) evaluate their measurements to a transport of 14.7 Sv, highly energetic and with strong variations on various timescales. The modelled transports range between 11.1 and 13.8 Sv showing a strengthening with increasing model resolution (Table 8). At 26.5°N, where observational measurements derived from RAPID (Rayner et al., 2011; Smeed et al., 2017) provide a reference value for the maximum overturning transport of 17.7 Sv for the period 2005 to 2009, the simulated transports are 3 to 4.5 Sv weaker which is a typical behaviour of models with limited horizontal resolution in the North Atlantic, not being able to correctly resolve the mesoscale and consequently the correct characteristic and spreading of lower North Atlantic Deep Water

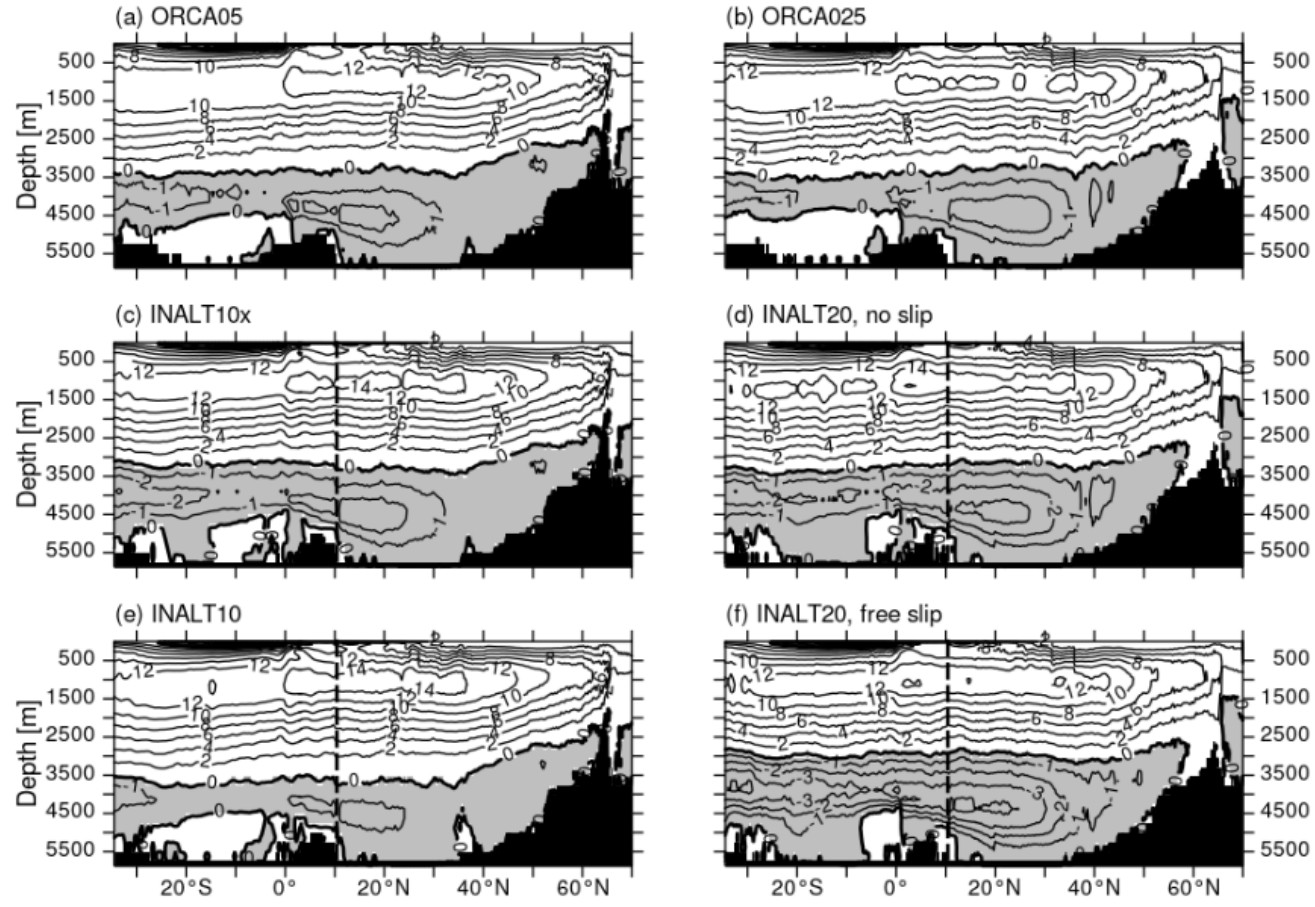

**Figure 11.** Mean AMOC [in Sv] for the period 2005-2009 from the hindcast experiments within the different configurations (a) ORCA05, (b) ORCA025, (c) INALT10x, INALT20 with no slip (d) and free slip (f) lateral boundary condition and (e) INALT10. The dashed lines in (c) to (f) indicate the northern boundary of the nested regions. Countour intervals are 1 Sv for negative (grey shaded) and 2 Sv for positive (white) transports.

from its formation region in the Nordic Seas through Denmark Strait into the subpolar North Atlantic (Behrens et al., 2013; Böning et al., 2016; Behrens et al., 2017).

The temporal evolution of the AMOC strength (Fig. 12) shows an admittedly weak but stable AMOC with long term trends all well below ±1 Sv per decade. In the North Atlantic at 26.5° N interannual correlations are above 0.86 among the simulations in the different configurations while in the South, at 34.5° S, the correlations are between 0.4 and 0.7 except for ORCA05 and INALT10 with a correlation of 0.8 (see Table 9 for all correlation coefficients). The high correlations in the North are an expression for the determination of the interannual variability by the applied forcing, that is the same for all simulations. Here, none of the nested configurations is represented at eddying resolution. Consequently, the particular nested configurations show a good correlation with the corresponding un-nested configurations with the same horizontal resolution (INALT10(x) and

**Table 8.** MOC transports at 34.5° S | 26.5° N [in Sv] averaged over the full integration period (1958 to 2009) and the respective standard deviations (std) on interannual timescales and for the period 2005-2009 compared to observations from Meinen et al. (2018) for 34.5°S and RAPID at 26.5°N (http://www.rapid.ac.uk/rapidmoc), respectively.

|  | 1958-2009 (mean ± std) | 2005-2009 |
|---|---|---|
| Observations |  | 14.7 | 17.7 |
| ORCA05 | 11.2 ± 0.56 | 12.4 ± 0.79 | 11.1 | 13.2 |
| ORCA025 | 12.9 ± 0.79 | 14.6 ± 0.56 | 12.8 | 13.9 |
| INALT10 | 12.6 ± 0.63 | 14.0 ± 0.76 | 12.4 | 14.7 |
| INALT10x | 12.7 ± 0.57 | 14.1 ± 0.73 | 12.4 | 14.5 |
| INALT20 NS_RW | 13.8 ± 0.61 | 14.9 ± 0.70 | 13.8 | 14.2 |
| INALT20 FS_RW |  | 12.2 | 13.3 |
| INALT20 FS_AW |  | 12.1 | 13.5 |
| INALT20 FS_PW |  | 12.2 | 13.3 |

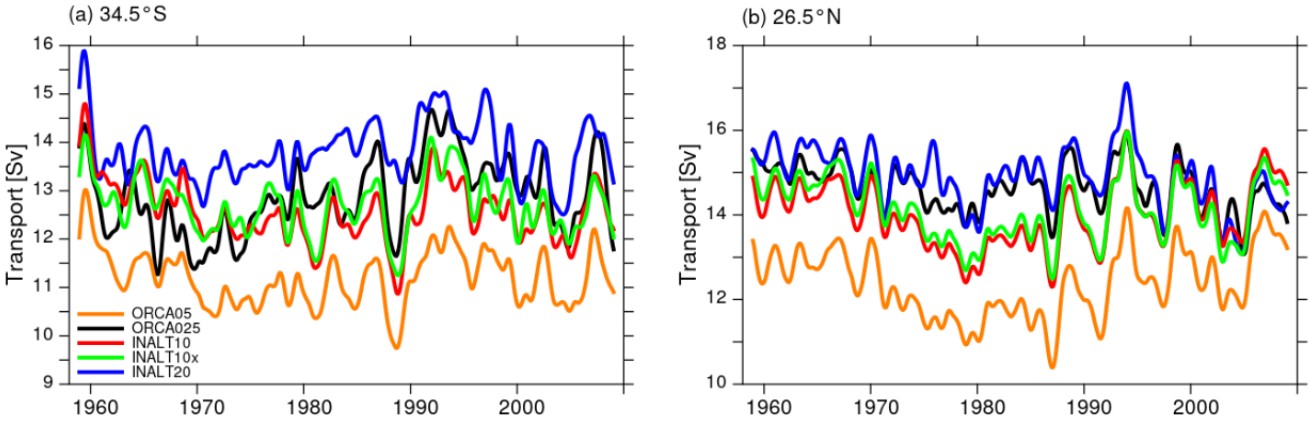

**Figure 12.** Timeseries of AMOC transports [in Sv] at 34.5° S (a) and 26.5° N (b) from ORCA025 (black), INALT20 (blue), ORCA05 (orange), INALT10 (red) and INALT10x (green)

ORCA05 as well as INALT20 and ORCA025). In the South, the correlations among the model solutions are reduced due to the partly resolved mesoscale and the associated non-linearities. This is highlighted by the difference in comparing INALT10 to ORCA05 and INALT20 to ORCA025. The nearly fully resolved mesoscale in INALT20 effectively reduces the correlation (although still significant) to the solution from the ORCA025 simulation, while INALT10 does not resolve the mesoscale south of 50° S and thus is correlated to the ORCA05 solution with a correlation coefficient of 0.92.

**Table 9.** Correlations of AMOC time series at 34.5° | 26.5° N based on annual averages. Significant correlations at 99% confidence level are given in bold.

|          | ORCA025       | INALT10       | INALT10x      | INALT20       |
|----------|---------------|---------------|---------------|---------------|
| ORCA05   | **0.75** \| 0.61 | **0.92** \| **0.98** | **0.88** \| **0.97** | 0.68 \| 0.56 |
| ORCA025  |               | **0.71** \| 0.67 | **0.78** \| **0.71** | **0.71** \| **0.79** |
| INALT10  |               |               | **0.85** \| **0.99** | 0.66 \| 0.65 |
| INALT10x |               |               |               | **0.69** \| **0.69** |

## 3.4 The Agulhas Current and the Agulhas Undercurrent

The AC has been observed in the early 2010s during the Agulhas Current Time-Series Experiment (ACT) utilizing a mooring array, starting between Port Alfred and East London and oriented perpendicular to the coast (Beal et al., 2015) at around 34° S. The resulting mean velocity structure across the section (Fig. 13(f)) shows the southwestward flowing AC being marked by a surface intensified, near coastal current with mean velocities up to roughly 1.5 m s$^{-1}$. The current's width is about 220 km. Below the AC, the AUC transports waters northeastward along the continental slope (Beal and Elipot, 2016; Beal et al., 2015).

All configurations simulate a surface intensified AC with a typical v-shaped pattern. The higher the resolution, the more asymmetric the structure, leading to maximum velocities closer to the coast. The width of the current, as seen in Fig. 13 or calculated according to Beal and Elipot (2016) (see Table 10 for all values) is (except for ORCA05) about 20-30 km wider than in observations. Despite the broader current structure, transports are slightly smaller due to weaker surface velocities and the current shows a slightly shallower depth expression. However, if calculated over a 5-year timeframe and fixed reference frame (T$_{box}$ according to Beal and Elipot (2016)), the eddy-rich configurations (INALT10(x) and INALT20) simulate transports within 5-10% of the observed value. The same is true for the representation of the short-term variability provided by the 5-daily standard deviations. Given this prominent short-term variability and interannual fluctuations (see below), the small discrepancy between the eddy-rich configurations and observations is expected. This is also demonstrated by the fact that INALT10 and INALT10x, although simulated at same resolution and parameters, show different transports. There is no clear resolution dependence of the transport. INALT60, however, simulates a higher transport, that is certainly not a representative value because of the limited averaging period and the fact that the model is in a different regime compared to the reference experiments (results are from years eight to ten in the spin-up experiments; see also Fig. 14). This is supported by the higher transport in INALT20 for the short period in the spin-up experiment.

Natal Pulses, that episodically deflect the AC from the coast, are simulated in all eddying configurations, which leads to the strong variations of the AC. With respect to the observational reference, one has to note that in the observed years at the ACT array the occurence of Natal Pulses has been very irregular (4 in 2010 and none in 2011 to 2013). Based on SST imagery, for the period 2004 to 2011, 1.6 Pulses per year are found (Rouault and Penven, 2011), while a longer timeseries covering the satellite period from 1993 to 2012 shows an increase of Natal Pulses per year from 1.3 before 2001 to 2.2 afterwards (Krug and Tournadre, 2012). The eddy-rich reference configurations range between 1.2 and 1.6 Natal Pulses per year in the period 2005 to 2009, that were defined as a deflection of the maximum velocity in the AC by more than two standard deviations off-shore (following Biastoch et al., 2018).

The representation of the AUC is clearly dependent on the horizontal grid resolution. Compared to the observed structure (Fig. 13(f)), the non-eddying simulation in ORCA05 (Fig. 13(a)) completely lacks an AUC, while the core of the AC is further off-shore and weaker, than the observed. It also features the strongest off-shore recirculation. ORCA025 (Fig. 13(b)) is already able to simulate an, admittedly weak, AUC and still shows this relatively weak and off-shore core of the AC, although more surface-confined. Increasing the resolution leads to an increase in the strength of the Undercurrent along with a broadening of the AC. All calcuated transports for the eddy-rich configurations are generally too low by 40-50% compared to the observations

**Table 10.** Mean (2005-2009) transports [in Sv] and standard deviations based on 5-daily averagres at the ACT section, current widths [in km] and number of Natal pulses (NP) per year in the corresponding period. Observational numbers are based on Beal and Elipot (2016). The Agulhas Undercurrent transport is given as the full depth integrated northeastward transport from the coast to 200 km off-shore and for the water column below 1000 m. * Years of the lower two rows refer to the spinup periods.

| | AC [Sv] | AC width [km] | AUC [Sv] (full depth \| below 1000 m) | NP [year$^{-1}$] |
|---|---|---|---|---|
| Observations | 77±31 | 219 | 10.7±9.8 \| 8.6±6.7 | 1.3 |
| ORCA05 | 63.3±3.6 | 203 | 0.1±0.1 \| 0 | 0 |
| ORCA025 | 61.3±22.1 | 246 | 4.2±5.7 \| 3.2±4.3 | 0.4 |
| INALT10 | 70.0±28.9 | 236 | 6.0±6.4\| 4.8±5.1 | 1.6 |
| INALT10x | 76.3±32.4 | 247 | 6.1±8.1 \| 5.3±6.3 | 1.2 |
| INALT20 NS_RW | 76.3±30.7 | 239 | 6.9±8.3 \| 5.7±6.4 | 1.4 |
| INALT20 FS_RW | 73.6±23.8 | 237 | 6.6±6.2 \| 5.7±5.0 | 1.4 |
| INALT20 FS_AW | 105.3±89.1 | 205 | 17.4±30.6 \| 14.9±23.5 | 2.2 |
| INALT20 FS_PW | 86.5±29.5 | 250 | 7.3±8.2 \| 6.3±6.5 | 1.4 |
| INALT20 (1987-1989)* | 86.3±32.6 | 234 | 6.2±8.1 \| 5.2±6.5 | 2.0 |
| INALT60 (1987-1989)* | 80.8±28.4 | 203 | 7.1±10.3 \| 5.6±7.7 | 0.3 |

(Beal and Elipot, 2016). This is dependent on potential depth restrictions of the calculation (full depth vs. below 1000 m), e.g. to separate the undercurrent structure from the countercurrent appearing during the passing of Natal Pulses (Biastoch et al., 2009a). It has to be noted that simulated (full depth) transports agree better with the 4.5 ± 5.2 Sv reported by Beal (2009), although the latitude at which the transport is calculated and the period taken into account differs between the modelled solutions and observations.

For both, the AC and AUC structures, the lateral boundary conditions have only limited impact (Fig. 14(d)). Although the Undercurrent in the free slip configuration appears closer to the continental slope, transports do not show a significant difference compared to the reference configuration. However, the influence of the ocean currents on the wind stress calculation has a large impact (Fig. 14(e) and (f)). The South East trade winds in this region typically contribute an additional Ekman component in the direction of the southward flowing surface current. The increase in AUC transport can be seen as a direct consequence of the enhanced AC.

Except for ORCA05, all configurations simulate a substantial interannual variability of the AC and AUC transports (Fig. 15). Rather than a direct relation to the atmospheric forcing, this is a result of the mesoscale variability and Natal Pulses (non-linearity). Earlier simulations with 1/10° configurations under climatological (repeated-year) surface forcing also exhibit a similar degree of interannual varaibility in comparison to their interannual-varying forced counterparts (Biastoch et al., 2009b; Loveday et al., 2014). It is therefore expected that the individual configurations do not correlate with each other nor are correlated with the observations. However, all configurations experience a long-term decline of the AC transport that was

already reported by Biastoch et al. (2009b). In contrast to sensitivity experiments presented by Loveday et al. (2014), who show an increase in the Agulhas Current transport in response to increasing trade winds in the Indian Ocean and a symmetrical response, for the tropical gyre circulation, no significant trend in the trade winds in the Indian Ocean can be found in the used CORE forcing. An evaluation of the declining AC trend will be performed elsewhere.

The common wind forcing is also responsible for the seasonal cycle (Fig. 15 (b)), robustly exhibiting low values in austral winter and high values in austral summer in all configurations and consistent with the observations (Krug and Tournadre, 2012; Beal et al., 2015). For the AUC, both amplitudes of short-term and interannual variability are larger than the mean value, which is a results of Natal Pulses (Biastoch et al., 2009a). In agreement with observations, none of the configurations simulates a seasonal cycle of the AUC transport (not shown).

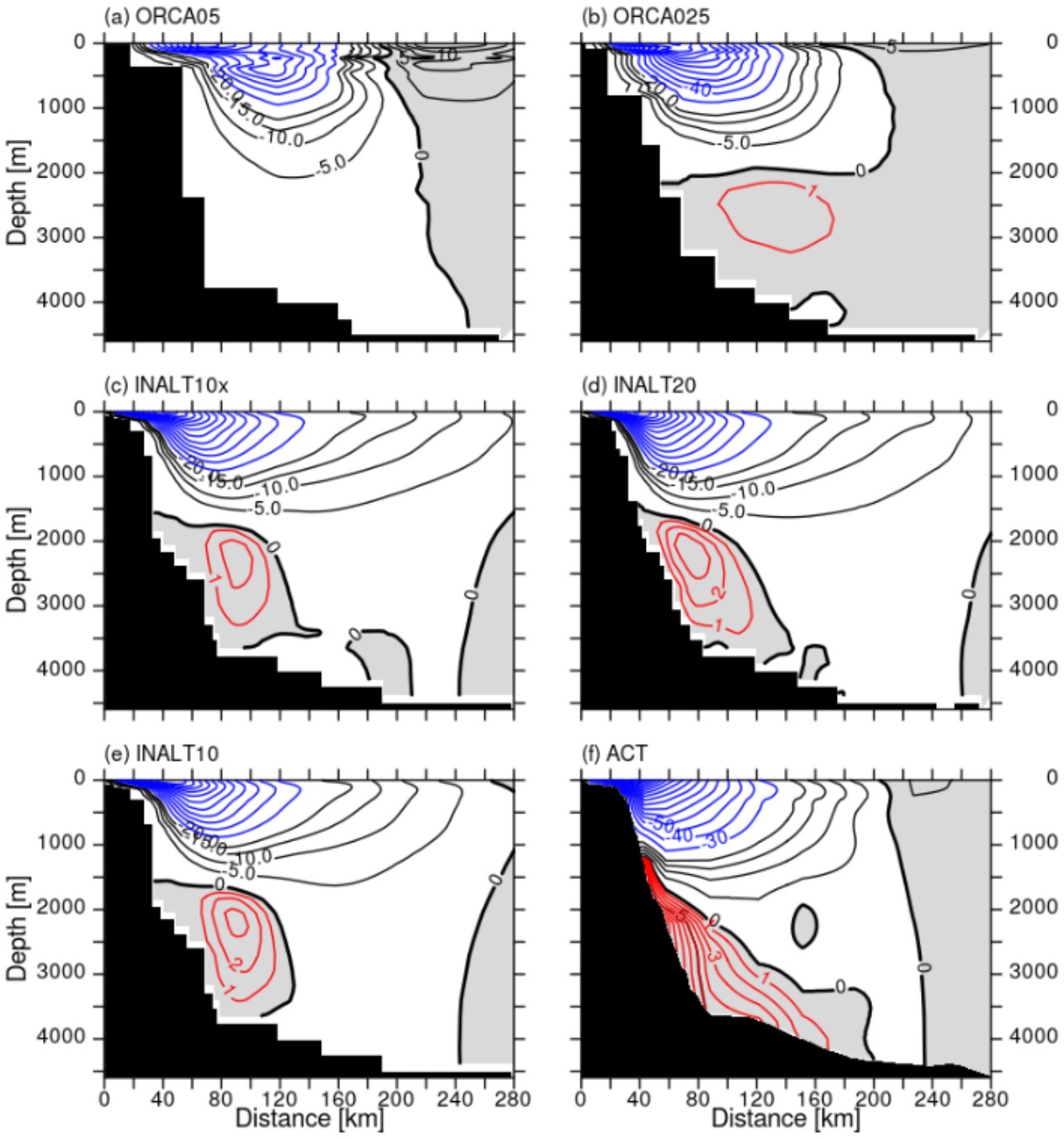

**Figure 13.** Velocity section perpendicular to the ACT section at ∼34° S, for the simulations (a-e) averaged from 2005-2009 and for the observations (f) averaged between 17 April 2010 and 19 February 2013. Southwestward/negative velocities (AC) are shaded in white, northeastward/positive (AUC) in grey, contoured are intervals of 1 cm s$^{-1}$ in red, 5 cm s$^{-1}$ in black and 10 cm s$^{-1}$ in blue.

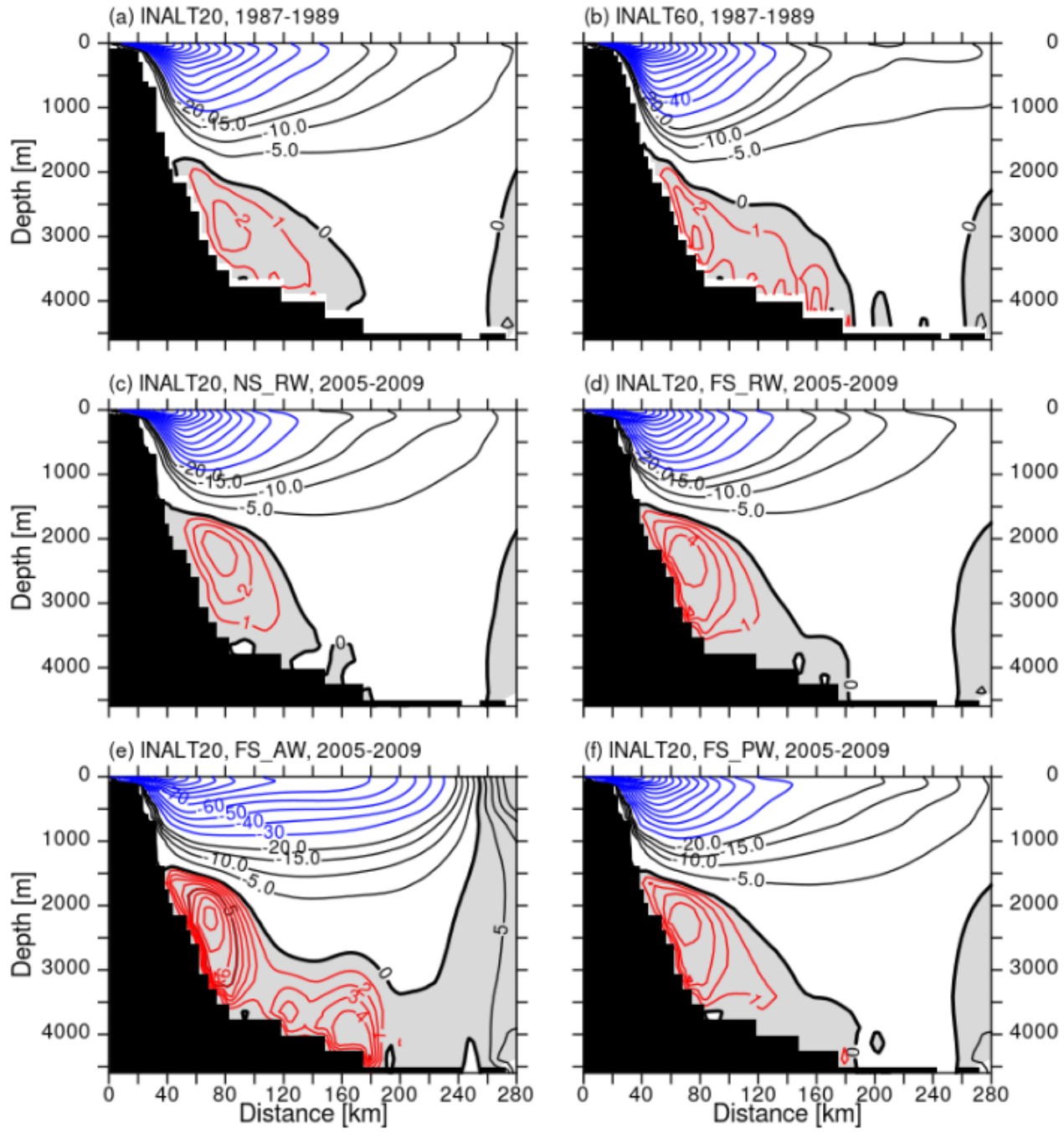

**Figure 14.** As Figure 13 but (a) INALT20 and (b) INALT60, averaged over years 8 to 10 (1987-1989) of the spinup experiments and the for the sensitivity experiments in INALT20 averaged over the period 2000-2009 (c to f)

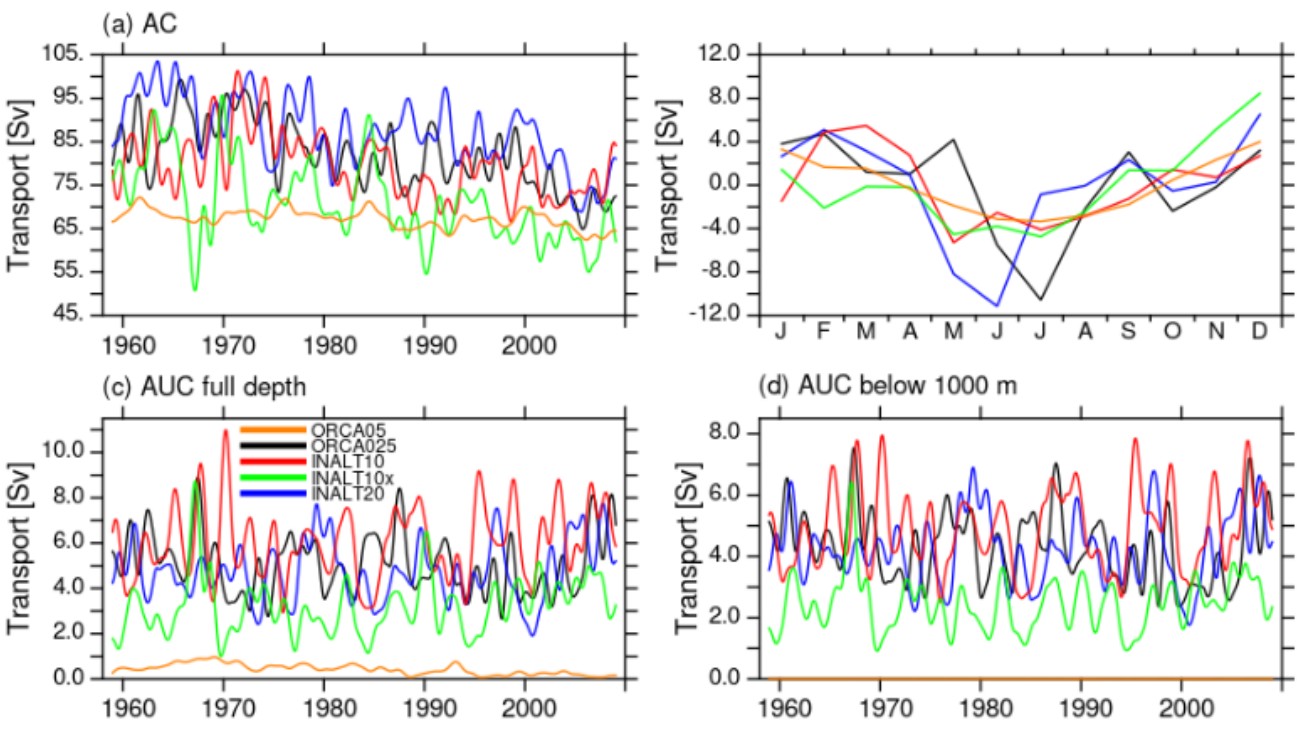

**Figure 15.** Agulhas Current transport timeseries (a) and annual cycle of transport anomalies (b) [in Sv]. Agulhas Undercurrent transport [in Sv] defined as the full depth integrated (c) and below 1000 m depth (d) northeastward velocity across the ACT section between the coast and 200 km off-shore.

**Table 11.** Mean (1995-2005) transports [in Sv] and linear trends from 1965-2000 [in Sv per decade] for the Agulas Current (AC), Agulhas Return Current (AR) and Agulhas Leakage (AL) as simulated in the Lagrangian experiments.

|          | AC          | AR          | AL         |
|----------|-------------|-------------|------------|
| ORCA05   | 61.3 \| -1.3 | 23.6 \| -2.3 | 35.5 \| 1.1 |
| ORCA025  | 60.8 \| -1.7 | 37.0 \| -2.6 | 22.0 \| 1.1 |
| INALT10  | 56.8 \| -2.1 | 39.0 \| -3.1 | 13.9 \| 1.2 |
| INALT10x | 57.0 \| -1.8 | 41.0 \| -3.1 | 12.2 \| 1.3 |
| INALT20  | 60.3 \| -1.9 | 42.0 \| -2.8 | 12.9 \| 1.1 |

## 3.5 Agulhas leakage

The flow from the Indian Ocean to the Atlantic Ocean through the Agulhas system is given by the Agulhas leakage. Owing to high spatio-temporal variability, it is typically calculated in a Lagrangian way (e.g., Biastoch et al., 2009b). Virtual particles are released at a section across the AC at 32° S, continously over one year and advected forward using the 5-daily time-varying three-dimensional flow field over a total of 5 years. Each particle is tagged with an initial transport that is kept constant during the integration. The transports of the particles that cross the Good Hope line (yellow in Fig. 16) are summed up and referred to as Agulhas leakage, those crossing the 35° E meridian are defined as the Agulhas Return Current. This is done for every single year of each model experiment, leading to annual time series of the AC, the Agulhas Return Current and Agulhas leakage transports. This method is a well-established methodology for ocean models (Durgadoo et al., 2013; Biastoch et al., 2009b, 2015), mimicking the sparse observations from surface and intermediate depth floats (Richardson, 2007).

In contrast to the ACT section described above, the AC here exhibits much more similarities across all experiments, with only INALT10 being generally weaker. The reason for this behaviour is that at this latitude (32° S) the AC is more stable and linear (Bryden et al., 2005). Further south at ACT, a meandering of the current takes place, together with local recirculations.

ORCA025, INALT10x and INALT20 are subject to a comparable decline of the AC transport of around 1.8 Sv per decade while the trend in ORCA05 is lower and in INALT10 it is slightly higher (Table 11).

Excluding ORCA05, 60% (ORCA025) to 70% (INALT configurations) of the initial AC transport retroflects and flows back into the Indian Ocean, along the Agulhas Return Current. ORCA05 (and to a certain degree also ORCA025) feature a too strong Agulhas leakage compared to the canonical number of 15 Sv in the upper 1000 m estimated by Richardson (2007). This resolution dependence of Agulhas leakage is well documented and a result of adequately represented mesoscale processes (e.g. Durgadoo et al., 2013). The INALT configurations simulate an Agulhas leakage in the range of 12 to 14 Sv, and are thus slightly lower compared to the observational estimate. Quite robust among all model configurations is the upward trend of Agulhas leakage between 1960s and 2000s, mainly a result of the strengthening of the Southern Hemisphere westerlies in the common wind forcing (Durgadoo et al., 2013). The simulated trend of 1.1-1.3 Sv per decade is in the range of the observationally-based index provided through east-west gradients of sea surface temperature (Biastoch et al., 2015).

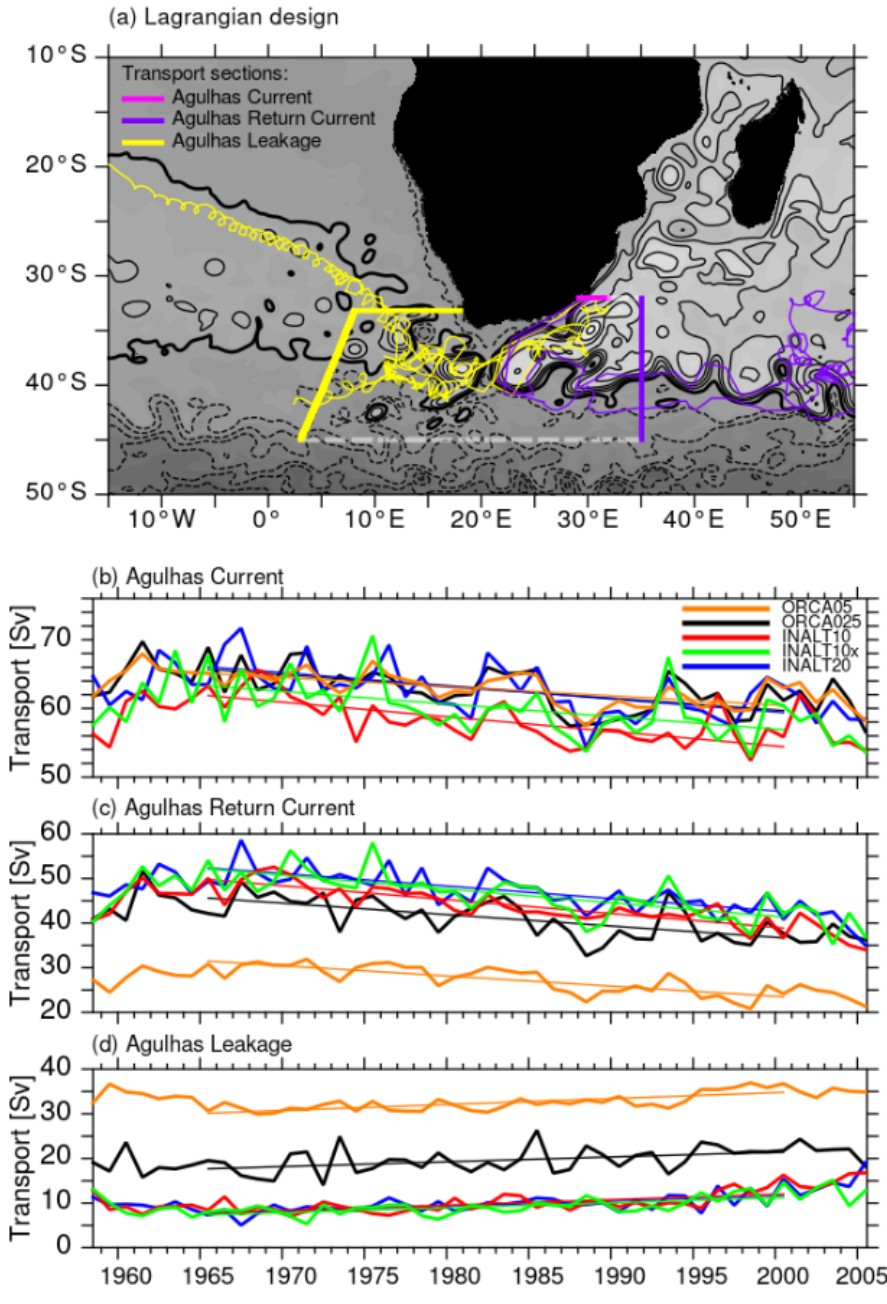

**Figure 16.** Lagrangian design (a) and transport time series [in Sv] of the Agulhas Current ((b); transport across the magenta line in (a)), Return Current ((c), transport across the purple line in (a)) and leakage ((d), transport across the yellow line in (a)) from the Lagrangian experiments. The shading and contours in (a) are a snapshot of SSH as simulated in INALT20. The yellow and purple trajectories indicate exemplary paths from the release to the sampling sections.

# 4 Conclusions

A series of nested global ocean model configurations, called the INALT family, has been established with horizontal resolutions of 1/10°, 1/20° and 1/60° in the South Atlantic and western Indian oceans. It has been shown that already at 1/10° horizontal resolution it is possible to simulate the eddy-generating instability processes in the source regions of the Agulhas Current and in the greater Agulhas Current system. An open question remains, whether 1/10° is already sufficient to represent the full range of mesoscale processes. Given the decrease of the baroclinic Rossby radius towards higher latitudes, it follows that at least 1/20°, probably even higher, resolution is required to represent mesoscale processes in the Southern Ocean.

The use of well-established global configurations as hosts for the nested configurations provides an important prerequisite. The corresponding configurations, ORCA05 and ORCA025, determine the embedment of the large-scale transports such as the Atlantic meridional ocerturning circulation, the supergyre and the Antarctic Circumpolar Current. It is demonstrated that the temporal variability of these measures in the hosts of the nested configurations is correlated with the respective un-nested configurations on interannual to decadal timescales. Nevertheless, the mesoscale dynamics in the nests also feed back to the global scale.

The representation of mesoscale variability in the region covered by the high-resolution nests improves with increasing horizontal resolution. In mid-latitudes, the configurations at 1/10°, INALT10 and INALT10x, already capture most of the observed variability, while at higher latitudes, INALT20 at 1/20° resolution shows a more realistically simulated variability, not only due to the high resolution in the nested region but also due to the eddying resolution on the host grid.

Resolving mesoscale processes leads to a more realistic representation of the highly variable Agulhas Current system. Upstream the Agulhas Current, the mean barotropic transport through the Mozambique Channel is comparable to observations in the un-nested configurations ORCA05 and ORCA025 at the expense of non-existent mesoscale eddies. The high-resolution nests properly simulate the mesoscale variability, while their transports are significantly weaker. This points to an under-representation of the eddy transport in the Mozambique Channel. Despite the offset in the mean, all model configurations reproduce the observed seasonal cycle in the transport through the Mozambique Channel and are significantly correlated with observations at monthly timescales.

The Agulhas Current transport itself increases with increasing resolution up to 1/10°, while beyond it slightly decreases. The transport of Agulhas Undercurrent, located on the continental slope below the Agulhas Current, further increases through all model resolutions. Both currents become more realistic in their mean transports and spatial structure in the eddy-rich configurations. Owing to the bathymetric impact on the undercurrent, INALT60 shows an additional level of improvement against INALT20. It was shown, that transport estimates and current structure realism not only depend on resolution but obviously also on details of the numerical settings in terms of boundary conditions. Altering lateral boundary conditions and the use of ocean currents in the wind stress calculation leads to a large range of simulated transports. In addition, strong intrinsic variations lead to interannual variations, preventing a proper comparison with the limited observational estimates of the currents. However, on longer timescales, the AC transports show comparable trends among the different configurations.

Downstream the Agulhas Current, where it separates into the Agulhas Return Current flowing back into the Indian Ocean and the Agulhas leakage that transports Indian Ocean waters into the Atlantic Ocean, all eddy-active configurations show a comparable behaviour. The majority of the transport (60 to 70%) retrocflects into the Indian Ocean, and only a minor portion finds its way into the Atlantic at comparable levels. The amount of Agulhas leakage simulated within the eddying configurations resemble the estimate from the sparse observations. Although the annual transports of the Agulhas Current, the Agulhas Return Current and the Agulhas leakage are naturally decorrelated among the different configurations because of the nonlinearity involved, on longer timescales and as a result of the common forcing, they show comparable trends.

In addition to the greater Agulhas Current system, the representation of the western boundary currents in the South Atlantic, the Malvinas and North Brazil Currents, show a general improvement with increasing model resolutions in structure and mean transports. The experiments also emphasize the importance of mesoscale processes in the Atlantic sector of the Southern Ocean on the representation of the Malvinas Current.

The resulting Atlantic meridional overturning circulation is robust concerning its structure among the different configurations with only the Antarctic Bottom Water cell showing a pronounced dependency on the resolution in the Atlantic and western Indian Ocean sectors of the Southern Ocean. The Antarctic Bottom Water cell is stronger and vertically more extended in INALT10x and INALT20 when compared to the other configurations. The mean transports at the meridional locations of the observational measurement sites SAMBA in the South and RAPID in the North Atlantic only show a slight increase with increasing resolution among the eddying configurations, whereas ORCA05 stands out with sigificantly weaker transports. Nevertheless, the observed mean transports are still underestimated. The mean transport of the Atlantic meridional overturning circulation consequently appears to be primarily dependent on driving factors in the North Atlantic, while its interannual variability is strongly forcing dependent.

Most of the mesoscale variability in the greater Agulhas Current region and the South Atlantic is simulated comparably at 1/10° and 1/20° resolution. The comparison between INALT10 and INALT10x can be used to isolate and study the Southern Ocean influence on the Agulhas dynamics. However, a weakness of ORCA05 providing the host for the two configurations at 1/10° remains: even in INALT10x the variability entering the South Atlantic through Drake Passage is absent, in contrast to INALT20, where the host configuration already simulates an eddying Antarctic Circumpolar Current. Comparing the eddy-poor ORCA05 and eddy-rich 1/10° configurations enables to specifically investigate the impact of resolving the mesoscale on a variety of physical processes.

The INALT family provides a consistent set of ocean model configurations. It allows to study a number of regional aspects, such as the cold vs. warm water route on the upper limb of the Atlantic meridional overturning circulation (Rühs et al., submitted) or the spreading of water masses in the South Atlantic (Tim et al., 2018). The range of resolution allows to address the impact of the mesoscale on the mean flow as well as mesoscale dynamics itself. INALT60 will be used to explore the way towards sub-mesoscale processes. The inclusion of the Southern Ocean sector, by comparing INALT10x and INALT10, allows to isolate the impact of Southern Ocean dynamics on the greater Agulhas Current system. INALT10 and INALT10x provide the oceanic basis for the coupling with an active atmosphere in the new Flexible Ocean Climate Infrastructure (FOCI, Matthes et al., in prep.).

*Code and data availability.* Data are available at https://data.geomar.de/thredds/catalog/open_access/schwarzkopf_et_al_2019_gmd/catalog. html. The model simulations are based on NEMO version 3.6 https://forge.ipsl.jussieu.fr/nemo/svn/NEMO/releases/release-3.6 (revision 6721). The AGRIF library is part of the NEMO release but maintained by INRIA. Documentation about the library itself as well as access to the code are available at agrif.imag.fr.

*Author contributions.* F.U.S. developed the family of INALT configurations, performed all experiments in INALT20(r) and INALT60 and conducted all analyses. F.U.S. and A.B. jointly wrote the manuscript. J.C. supported the model development as part of the NEMO system team. J.H. performed the model experiments in ORCA05, and INALT10(x), J.K.R. performed the model experiments in ORCA025. M.M.S. set up and supported the model environment for all experiments. R.S. contributed the sub-mesoscale part of the manuscript. All authors were involved at different stages of the establishment of the model family and contributed to the final state manuscript.

*Competing interests.* The authors declare that they have no conflict of interest.

*Acknowledgements.* This study project received funding from the German Federal Ministry of Education and Research (BMBF) of the SPACES-AGULHAS project, grant 03F0750A and SPACES-CASISAC grant 03F0796A. J.V.D. acknowledges funding from the Helmholtz-Gemeinschaft and the GEOMAR Helmholtz Centre for Ocean Research Kiel, grant IV014/GH018. All model simulations have been performed at the North-German Supercomputing Alliance (HLRN). We gratefully acknowledge the work by the NEMO system team and especially thank Rachid Benshila for his assistance during the model set up phase. We thank Tobias Schulzki for generating Figure 1. Our thanks also go to Dr. Jenny Ulgren and NIOZ Royal Netherlands Institute for Sea Research for prividing the observed Mozambique Channel transport time series (LOCO). The authors also thank two anonymous reviewers for investing their time in improving the manuscript.

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
