# Peer review of "The INALT\* family - a set of high-resolution nests for the Agulhas Current system within global NEMO ocean/sea-ice configurations"

_Geoscientific Model Development, 2018_

## Referee Comment (RC1) · Anonymous Referee #1 · 21 Mar 2019

This long manuscript presents a family of region ocean model configurations centered around the Agulhas Current System. These configurations are all based on the NEMO3.6-LIM2 ocean circulation and ice model. They are all nested into global models (based on ORCA05 or ORCA025 from the Drakkar project), use ETOPO1 or ETOPO2 as bottom topography and are forced by the COREv2 product by a bulk formula.

The configuration are parameters are detailled. The 1/10°, 1/20°, 1/60° resolution configurations are called INALT10, INALT20 and INALT60.

Several measures of model behaviour are provided: SSH variance, Transport of the ACC, Transport through the Mozambique Channel, Malvinas and North Brazil Current,

Merdional Overturning Circulation, The Agulhas Current and Undercurrent, Agulhas Leakage. A comparison with observations is given when possible shoiwng the abilities of this configurations.

Tests are made on grid resolution, lateral conditions of z-coordinate models on topography (free slip or non slip) and on the effect of ocean currents on wind stress.

In addition to a general assessment of this configurations the main message is that the dynamics of the Agulhas Current system depend on the representation of mesoscale processes with the Agulhas Current and the Agulhas Undercurrent transports increase with increasing resolution.

General Comments

This article is clear and well written (although a bit tedious). Figure quality is satisfactory.

Although the scientific message is relatively limited, this presentation in great lengths of these configurations is valuable.

The approach is conservative, using a model designed originally for large scale ocean circulation (NEMO), based on second order numerics, using coarse surface forcing (CORE) and Laplacian diffusion; but increasing the horizontal resolution.

It is notorious that modelling the Agulhas Current is a difficult exercice. The solution found here to harness the beast was to increase the resolution while keeping relatively large values of Laplacian diffusion and biharmonic viscosity (see Table 2). Combinations of biharmonic and Laplancian operators have been used in the past to stabilise the Gulf Stream (Chassignet and Garrafo, 2001), the Laplacian operator bringing a dissipation acting at all scales (Soufflet et al., 2016). Here, the value of Laplacian diffusion used for INALT10 (120 to 400 m2 s-1) is comparable to the Laplacian viscosity used by Chassignet and Boudra (1988): 330 m2 s-1 for a 40 km resolution experiment made in 1988.

The comparison of the large scale and regional circulation with observations is interesting, although long term trends such as in the Mozambique Channel Transport (Fig8 a) are a bit surprising.

There was a couple errors with the literature: Rouault et al. (2009) did not report a long term decline in AC transport. Loveday et al. (2014) did not report a decrease of the AC transport in reponse to decreasing trade winds.

In regard of these comments I recommend publication after minor corrections.

**Specific Comments**

Figure 2: It would be clearer if the names of the configurations could be seen directly on the zooms

Table 2: It got confused with the units of ahm0 [m2 s-1] while the values were typical of biharmonic viscosities. and ahtm [m4 s-1] while values were typical of Laplacian diffusions. I got also confused with |ahmm| > |ahm0| for ORCA05 while |ahmm| < |ahm0| for all the other runs. Why ? Please clarify.

Figure 9: I had doubts of the interest of the comparison of these meridional velocities so close to a current separation. We get here opposite currents while differences between the run could be small.

Page 29: Improper reference to Rouault et al. (2009)

Page 30: Improper reference to Loveday et al. (2014)

References

Chassignet, E. P., and Z. D. Garraffo (2001), Viscosity parameterization and the Gulf Stream separation, in From Stirring to Mixing in a StratifiedOcean: Proceedings of Aha Huliko, a Hawaiian Winter Workshop, Honolulu, Hawaii, pp. 37-41, Univ. of Hawaii at Manoa, Honolulu.

СЗ

Soufflet, Y., P. Marchesiello, F. Lemarie, J. Jouanno, X. Capet, L. Debreu, R. Benshila, On effective resolution in ocean models, 2016, Ocean Modelling, 98, pp.36-50

---

## Referee Comment (RC2) · Anonymous Referee #2 · 24 May 2019

This paper gives a very thorough description of a family of model configurations focussed on nesting in and understanding the Agulhas System and its impacts on the global circulation. It gives a well structured and informative summary of 7 different configurations at various resolutions, with bases in the ORCA family of models. The configurations share a common atmospheric forcing and vertical grid, and a series of sensitivity studies on resolution and lateral boundary conditions are performed, with comparisons to various observed datasets. Although the manuscript is lengthy, its structure and the details it gives with regards to the configurations and their parameters will make a valuable piece of reference material for future work using these, or other, configurations in the southern hemisphere. The tests using relative, partial and

absolute winds in particular, add to what is an active research topic in various modelling systems.

The section (3.1.1) on the submesoscale is a valuable contribution to the paper, and is informative and refreshing in its honesty about the challenges and opportunities in the high resolution simulation. With the Agulhas system being tricky to accurately represent in regional and global ocean models, the parameters given in table 2 reveal some of the ingredients to the success of these configurations.

In general the paper is very well written, and with the addition of a few references, and some explanation around the choice of the common atmospheric forcing and vertical grid, is ready for publication.

Specific Comments

Section 1. Line 24: When discussing the generation of Natal Pulses, I think it would be remiss not to mention the work of Tsugawa and Hasumi (2010)

Figure 2: Suggest adding the name of the configuration next to the relevant nest.

Section 2. Line 4: I believe that Cronin et. al. (2013) and Malan et. al. (2019) also make use of INALT01.

Section 2.3 I think it is important here to discuss the philosophy behind using COREv2 as the atmospheric forcing for all experiments. I understand the need for a common forcing across all experiments, although for the higher resolution configurations I believe that the resolution of CORE may be a limitation, as is acknowledged briefly in section 3.1.1 of the paper. Basically I do not think that there is anything wrong with the choice, but think that some discussion of the reasoning behind the approach, and possible pro's and con's, would be instructive, the same comment, to a lesser extent, is also valid for the use of the 46-level vertical grid for all the simulations (page 6, line 6).

Table 10

INALT20 and 60 show a very low number of Natal Pulses during their spin-up period. Why is this?

References: Cronin, M. F., Tozuka, T., Biastoch, A., Durgadoo, J. V, & Beal, L. M. (2013). Prevalence of strong bottom currents in the greater Agulhas system. Geophysical Research Letters, 40, 1772–1776. https://doi.org/10.1002/grl.50400 Malan, N., Durgadoo, J. V., Biastoch, A., Reason, C., & Hermes, J. (2019). Multidecadal wind variability drives temperature shifts on the Agulhas Bank. Journal of Geophysical Research: Oceans. https://doi.org/10.1029/2018jc014614 Tsugawa, M., & Hasumi, H. (2010). Generation and Growth Mechanism of the Natal Pulse. Journal of Physical Oceanography, 40(7), 1597–1612. https://doi.org/10.1175/2010JPO4347.1

---

## Author Comment (AC1) · 19 Jun 2019

We thank the two anonymous reviewers for the time they invested in reading and commenting on our manuscript and appreciate their patience to carefully go through it in its full length. In the following, Reviewer Comments are given in black, Author Comments in cyan and changes to the manuscript in italic with tracked changes (red refer to deletions; blue refer to additions), referring to the occurence in the original manuscript.

Additionally to the changes implemented upon suggestion from the two reviewers

(presented below) we want to rephrase one paragraph in the Conclusions:

page 37, line 5: *However, on longer timescales and as a result of the common forcing the Although the annual transports of the Agulhas Current, the Agulhas Return Current and the Agulhas leakage , show comparable trends are naturally decorrelated among the different configurations because of the nonlinearity involved, on longer timescales and as a result of the common forcing, they show comparable trends.*

We also updated the acknowledgements and some references and corrected a few typos.

**1 Anonymous Referee #1**

This long manuscript presents a family of region ocean model configurations centered around the Agulhas Current System. These configurations are all based on the NEMO3.6-LIM2 ocean circulation and ice model. They are all nested into global models (based on ORCA05 or ORCA025 from the Drakkar project), use ETOPO1 or ETOPO2 as bottom topography and are forced by the COREv2 product by a bulk formula.

The configuration are parameters are detailled. The $1/10°$, $1/20°$, $1/60°$ resolution configurations are called INALT10, INALT20 and INALT60.

Several measures of model behaviour are provided: SSH variance, Transport of the ACC, Transport through the Mozambique Channel, Malvinas and North Brazil Current, Merdional Overturning Circulation, The Agulhas Current and Undercurrent, Agulhas Leakage. A comparison with observations is given when possible shoiwng the abilities of this configurations.

Tests are made on grid resolution, lateral conditions of z-coordinate models on topography (free slip or non slip) and on the effect of ocean currents on wind stress.

In addition to a general assesment of this configurations the main message is that the dynamics of the Agulhas Current system depend on the representation of mesoscale processes with the Agulhas Current and the Agulhas Undercurrent transports increase with increasing resolution.

General Comments

This article is clear and well written (although a bit tedious). Figure quality is satisfactory.

Although the scientific message is relatively limited, this presentation in great lengths of these configurations is valuable.

The approach is conservative, using a model designed originaly for large scale ocean circulation (NEMO), based on second order numerics, using coarse surface forcing (CORE) and Laplacian diffusion; but increasing the horizontal resolution.

It is notorious that modelling the Agulhas Current is a difficult exercice. The solution found here to harness the beast was to increase the resolution while keeping relatively large values of Laplacian diffusion and biharmonic viscosity (see Table 2). Combinations of biharmonic and Laplancian operators have been used in the past to stabilise the Gulf Stream (Chassignet and Garrafo, 2001), the Laplacian operator bringing a dissipation acting at all scales (Soufflet et al., 2016). Here, the value of Laplacian diffusion used for INALT10 (120 to 400 m2 s-1) is comparable to the Laplacian viscosity used by Chassignet and Boudra (1988): 330 m2 s-1 for a 40 km resolution experiment made in 1988.

The comparison of the large scale and regional circulation with observations is interesting, although long term trends such as in the Mozambique Channel Transport (Fig8

a) are a bit surprising.

This kind of long-term trend in modelled Mozambique Channel transports has already been described and attributed to trends in wind stress and wind stress curl over the Indian Ocean (e.g. DiMarco et al. 2002). We added this reference to the manuscript.

page 18, line 29: *We conclude that Consequently, this trend is likely wind-related and rather not an effect of model drift but likely to be related to trends in wind stress and wind-stress curl over the Indian Ocean (DiMarco etal. 2002)*

*Reference: DiMarco, S. F., Chapman, P., Nowlin, W. D., Hacker, P., Donohue, K., Luther, M., Johnson, G. C., and Toole, J.: Volume transport and property distributions of the Mozambique Channel, Deep Sea Research II, 49, 2002.*

There was a couple errors with the literature: Rouault et al. (2009) did not report a long term decline in AC transport. Loveday et al. (2014) did not report a decrease of the AC transport in response to decreasing trade winds. (Page 29: Improper reference to Rouault et al. (2009); Page 30: Improper reference to Loveday et al. (2014))

We thank the referee for carefully checking the references. Rouault et al. (2009) reported an increase in AC transport along with a warming in the Agulhas Current region. We removed this reference. Loveday et al. (2014) did not report a decrease of the AC transport in response to decreasing trade winds but attributed an increase in AC transport (as consequence of stronger flow in both the tropical and subtropical gyres) to an increase in the trade winds. For the tropical gyre, they state a symmetrical response when the trade winds weaken. We rephrased this part of the text as follows:

page 29, line 16: *However, all configurations experience a long-term decline of the AC transport that was already reported by Biastoch et al. 2009band Rouault et al.*

*2009. However, in . In contrast to sensitivity experiments presented by Loveday et al. (2014), who show the decrease of the AC an increase in the Agulhas Current transport in response to decreasing increasing trade winds in the Indian Ocean , and a symmetrical response, for the tropical gyre circulation, no significant trend in the trade winds in the Indian Ocean can be found in the used CORE forcing. An evaluation of the declining AC trend will be performed elsewhere.*

In regard of these comments I recommend publication after minor corrections.

Specific Comments

Figure 2: It would be clearer if the names of the configurations could be seen directly on the zooms
We added the names of the nested configurations to Figure 2.

Table 2: It got confused with the units of ahm0 [m2 s-1] while the values were typical of biharmonic viscosities. and ahtm [m4 s-1] while values were typical of Laplacian diffusions.

Thank you for pointing this out - the units for ahm and aht indeed got mixed up. We corrected the units in the manuscript accordingly. (see below)

I got also confused with |ahmm| > |ahm0| for ORCA05 while |ahmm|<|ahm0| for all the other runs. Why ? Please clarify.
We appreciate this comment very much. ahmm and atmm are actually inactive in the experiments we use here. We therefore exclude these values from Table 2.

**Table 2.** *Resolution dependent parameters for the different configurations: eddy induced velocity coefficient (aeiv0 [$m^2 s^{-1}$]); horizontal eddy diffusivity coefficients for tracer (aht0 [$m^2 s^{-1}$]) and momentum (ahm0 [$m^{24} s^{-1}$]) are nominally set for the maximum grid size (which is at the equator except for the nest grids in INALT20r and INALT60 that do not include the equator at high resolution) and scaled with the poleward decrease of the grids sizes; their upper limits are given by ahtm and ahmm (both in $m^4 s^{-1}$) respectively; lateral mixing coefficient in the bottom boundary layer (ahtbbl [$m^2 s^{-1}$]); sponge coefficients for tracer and dynamics (sponge [$m^2 s^{-1}$]). Note that ORCA05 also represents the values for the host grid in INALT10 and INALT10x, ORCA025 those for the host grid in INALT20, INALT20r and INALT60. The settings for the first nest in INALT60, are identical to those given for INALT20(r).*

| | aeiv0 | aht0 | ahtm | ahm0 | ahmm | ahtbbl | sponge |
|---|---|---|---|---|---|---|---|
| ORCA05 | 1000 | 600 | 2000 | $-6.0\times10^{11}$  $-6\times10^{11}$ | $-1\times10^{12}$ | 1000 | - |
| ORCA025 | - | 300 | 300 | $-1.5\times10^{11}$ | $-1\times10^{11}$ | 1000 | - |
| INALT10(x) | 0 | 120 | 400 | $-2.4\times10^{10}$ | $-8\times10^{9}$ | 100 | 2700 |
| INALT20(r) | - | 60 | 60 | $-6.0\times10^{9}$  $-6\times10^{9}$ | $-8\times10^{8}$ | 40 | 600 |
| INALT60 | - | 20 | 20 | $-6.7\times10^{8}$ | $-3.7\times10^{7}$ | 4.444 | 200 |

Figure 9: I had doubts of the interest of the comparison of these meridional velocities so close to a current separation. We get here opposite currents while differences between the run could be small.

The reason for the choice of this location was the comparability to observations provided by Artana et al. (2018). To verify the results presented in the manuscript, we further analyzed the velocity fields around the Malvinas Current (MC) as represented by INALT10 and INALT10x. INALT10 shows a generally weaker mean (2000-2009) circulation pattern feeding the MC (Fig. R1a; the thin horizontal line indicates the southern boundary of the nested region in INALT10) when compared to INALT10x (b).

Zonal sections of meridional velocities across the MC at three different latitudes (red, blue and green lines in a and b) south of the section provided in the main manuscript (thick black line in a and b) and the corresponding MC transports (see labels in Fig. R1c to h; volume transports are calculated as the integrated northward velocities from the coast to the longitude marked by the vertical thin lines in c to h) clearly show a weaker expression of the MC in INALT10 than in INALT10x, independently from the latitudinal position. We re-introduced a paragraph (that somehow accidentally got lost during the final compilation of the submitted manuscript) discussing the choice of this section and the results from our experiments.

page 19, line17: *The MC originates in the Drake Passage as the northern branch of the ACC, turning northward along the South American continental slope into the Atlantic Ocean. It consists of multiple jets (Piola et al., 2013) that align towards the north into a coherent current at 41° S whose mean structure and temporal variability have been assessed based on mooring data and satellite altimetry (Artana et al., 2018). Here, a comparison with the mean structure of the Malvinas Current as simulated in the different configurations is given (Fig. 9). ORCA05 and INALT10 both lack a MC at 41° S, due to a recirculation towards the East loacted further south that can be attributed to the missing resolution in the ACC and a consequently weaker MC. The configurations covering the entrance of the ACC into the Atlantic basin at eddying resolution (ORCA025, INALT10x and INALT20) all show a wide band of northward surface intensified velocities with a core located between 70 km and 80 km off-coast. The mean transports increase with increasing resolution from 25 Sv in ORCA025 to 37 Sv in INALT20 (see Table 7 for all mean transports). The latter well fits the observed transport, although featuring weaker velocities over a wider current core than in observations (Artana et al. 2018, their Fig. 5 (a)).*

**2 Anonymous Referee #2**

This paper gives a very thorough description of a family of model configurations focussed on nesting in and understanding the Agulhas System and its impacts on the global circulation. It gives a well structured and informative summary of 7 different configurations at various resolutions, with bases in the ORCA family of models. The configurations share a common atmospheric forcing and vertical grid, and a series of sensitivity studies on resolution and lateral boundary conditions are performed, with comparisons to various observed datasets. Although the manuscript is lengthy, its structure and the details it gives with regards to the configurations and their parameters will make a valuable piece of reference material for future work using these, or other, configurations in the southern hemisphere. The tests using relative, partial and absolute winds in particular, add to what is an active research topic in various modelling systems.

The section (3.1.1) on the submesoscale is a valuable contribution to the paper, and is informative and refreshing in its honesty about the challenges and opportunities in the high resolution simulation.

We thank the reviewer for honoring the outlook on our upcoming work with INALT60. We would like to introduce some minor changes to this section to be more precise in the description of te applied advection and diffusion schemes.

page 16, line 2: *INALT60 with horizontal grid spacing below 2 km provides the necessary horizontal resolution to resolve scales down to 10 km. In addition to the horizontal resolution, the vertical grid, the spatial and temporal resolution of the atmospheric forcing and the diffusion* parameters setup *strongly control the simulation of the smaller scale flows in the model. Developments to improve the configuration in the future are ongoing and are presented by Schubert et al. (under review). Here, a short outlook is*

*given.*

*To  adequately simulate the ocean currents potentially resolvable with the given horizontal grid, the vertical resolution needs at least to resolve the vertical structure of the corresponding horizontal flow which can be approximated by the baroclinic modes (Stewart et al. 2017). Based on hydrographic measurements, Stewart et al. (2017) provide a reference for the required vertical resolution to resolve the first, second and third baroclinic modes. A new vertical grid,  that shall resolve the third baroclinic mode, with 120 vertical levels  and a grid spacing of 1 m near the surface, 10 m at about 350 m depth and 100 m in the deep ocean is under development. A very high-vertical resolution  in the mixed layer is required, as sub-mesoscale currents mainly occur in boundary layers and are surface intensified (McWilliams, 2016). The mixed layer depth in the mid-latitudes regions of the storm tracks, such as the Agulhas region, is associated with a strong seasonal cycle. Weaker  wind-stress in summer leads to a thin mixed-layer of down to less than 20 m, while stronger  wind-stress in winter leads to a thick mixed-layers exceeding 150 m (de Boyer Montégut et al. 2014). To resolve smaller scale flows within the mixed-layer also during summer time, at least 10 vertical levels in the uppermost 20 m are required.*

*Near-surface sub-mesoscale features, resolvable with a 1/60° configuration, evolve at time scales less than a day (McWilliams, 2016).  Due to the sub-daily timescales, a realistic representation of the daily cycle in the forcing fields is necessary for simulating the sub-mesoscales properly. The CORE atmospheric forcing is given at relatively coarse resolution in space ($2° \times 2°$) and time (6 hourly for the highest resolved fields). For simulations with INALT60, the higher resolved JRA55-do forcing (Tsujino et al., 2018), which is associated with relatively higher horizontal ($0.5° \times 0.5°$) and temporal (3 hourly) resolutions, will be used in the future.*

*The modelled diffusion consists of Both the numerical diffusion associated with advection schemes for tracer and momentum and the explicit diffusion in the primitive*

*equations contribute to the model diffusion. So far, the same model diffusion for tracers (momentum) setup is used for INALT60 and INALT20: TVD (EEN) advection scheme , Laplacian (bi-Laplacian) scheme and Laplacian explicit diffusion with constant diffusion coefficient for tracers and Vector Invariant advection scheme with EEN vorticity formulation supported by bi-Laplacian explicit viscosity with a constant viscosity coefficient for momentum. The diffusion coefficient has been linearly (quadratically) scaled down with the grid spacing for tracers (momentum). The coefficient has thereby to be large enough to prevent the strongest simulated shears from numerical instability. A disadvantage of this method is that moderate shears that would not have lead to numerical instability are also damped. In the future, third-order upstream biased schemes (UBS, Webb et al., 1998; Farrow and Stevens, 1995; Madec and the NEMO team, 2014) for tracer and momentum will be used. These schemes are numerically diffusive enough at the grid-scale to inhibit numerical instabilities at the grid-scale. Explicit diffusion with UBS is only needed, if the numerical diffusion is not large enough to be realistic. The evaluation of the model performance with UBS and a respective validation against observations is presented by Schubert et al. (under review).*

With the Agulhas system being tricky to accurately represent in regional and global ocean models, the parameters given in table 2 reveal some of the ingredients to the success of these configurations.

In general the paper is very well written, and with the addition of a few references, and some explanation around the choice of the common atmospheric forcing and vertical grid, is ready for publication.

Specific Comments

Section 1. Line 24: When discussing the generation of Natal Pulses, I think it would be remiss not to mention the work of Tsugawa and Hasumi (2010)

We are aware that a significant number of modeling studies exist for the interpretation of Natal Pulses, however cannot provide an extensive discussion. Here we focus on the observational description (Schouten et al., 2002) and the influence on Agulhas rings (Rouault and Penven, 2011).

Figure 2: Suggest adding the name of the configuration next to the relevant nest.
We added the names of the nested configurations to Figure 2.

Section 2. Line 4: I believe that Cronin et. al. (2013) and Malan et. al. (2019) also make use of INALT01.
We are aware of these publictaions. The current list of publications didn't mean to be a complete list of publications using INALT01 but rather give examples. However, we added Cronin et. al. (2013) and Malan et. al. (2019).

page 4, line 3: *INALT01 has been widely used to understand AC dynamics (Rühs et al., 2017; Malan et al., 2018), the effect of the Southern Hemisphere wind systems on the AC (Durgadoo et al., 2013; Loveday et al., 2014), the impact of Agulhas leakage on the Atlantic circulation and hydrography (Biastoch et al., 2015; Lübbecke et al., 2015),* as well as local phenomena in the (greater) Agulhas region (Cronin et al., 2013; Malan et al., 2019) *and has been utilized for a range of interdisciplinary applications (Scussolini et al., 2013; Steinhardt et al., 2014; van Sebille et al., 2015).*

*References: Cronin, M. F., Tozuka, T., Biastoch, A., Durgadoo, J. V., and Beal, L. M.: Prevalence of strong bottom currents in the greater Agulhas system,Geophysical Research Letters, 40, 2013.*

*Malan, N., Durgadoo, J. V., Biastoch, ., Reason, C., and Hermes, J.: Multidecadal wind variability drives temperature shifts on the AgulhasBank, Journal of Geophysical Research, 124, 2019.*

Section 2.3 I think it is important here to discuss the philosophy behind using COREv2 as the atmospheric forcing for all experiments. I understand the need for a common forcing across all experiments, although for the higher resolution configurations I believe that the resolution of CORE may be a limitation, as is acknowledged briefly in section 3.1.1 of the paper. Basically I do not think that there is anything wrong with the choice, but think that some discussion of the reasoning behind the approach, and possible pro's and con's, would be instructive, the same comment, to a lesser extent, is also valid for the use of the 46-level vertical grid for all the simulations (page 6, line 6).

At the time, our experiments were performed, COREv2 was the most coherent dataset providing forcing data for a long hindcast period. We are aware, that not only the low horizontal resolution of the forcing but also the temporal resolution and the availability of some forcing variables only as climatological fields, might hinder the high resolution configurations in evolving their full vitality. With the recently released, and now ready-to-use, community based forcing JRA55-do (Tsujino et al., 2018) at higher spatial and temporal resolution, COREv2 will be replaced in current and future simulations. However, applying spatially higher-resolved forcing fields, especially to high-resolution models introduces new levels of complexity, e.g. one has to keep in mind, that the forcing itself mi ght include mesoscale features (e.g. in wind products derived from scatterometer) that not necessarily fit the mesoscale simulated by the high resolution ocean model below. We added the following discussion of the reasoning behind using COREv2 to the manuscript:

page 9, line 4: *All experiments are thus performed over the same integration length*

[Figure]

*. , utilizing the same atmospheric forcing. The comparisons between the different configurations therefore allow to isolate the impact of different resolutions. Despite COREv2 having a spatial resolution of 2° × 2°, being relatively coarse, it was chosen as the most coherent and robust dataset available at the time of the model integrations, to provide long (multi-decadal) hindcast simulations.*

Concerning the use of the 46-level vertical grid, we are aware of potential improvements for the simulation of mesoscale processes with a higher vertical resolution, when the horizontal resolution is high enough to simulate sub-mesoscale currents. This is briefly addressed in section 3.1.1. and analyzed in detail by Schubert et al. (under review). For our simulations up to 1/20° we do not see a significant improvement when going to higher vertical resolution for simulating the greater Agulhas System and its impact on large scale circulation. For example, the Agulhas (Under) Current variability is basically represented by the first baroclinic mode (Stewart et al. 2017), that is resolved by our 46 vertical level axis. However, when it comes to resolving sub-mesoscale processes (where INALT60 is close to), a higher vertical resolution, especially within the mixed-layer, has a stronger effect also on the representation of mesoscale processes. We added a short explanation of the choice of the vertical axis to the manuscript:

page 6, line 7: *All configurations share the same vertical grid with 46 z-levels varying in layer thickness from 6 m at the surface to 250 m in the deepest layers, resolving the first baroclinic mode (Stewart et al., 2017) which is needed for the representation of the major baroclinic currents. The same vertical grid has proven to be an appropriate choice for simulations with model configurations of up to 1/20° horizontal resolution (e.g. Böning et al., 2016; Behrens et al., 2017).*

Table 10

INALT20 and 60 show a very low number of Natal Pulses during their spin-up period. Why is this?

We thank the reviewer for pointing to the Natal Pulses. In the definition of a Natal Pulse, the reference mean location of the maximum AC was erronnously taken for the period 2009 - 2009 instead of 2005 - 2009 for the hindcast experiments and 1989 instead of 1987 - 1989 for the spin-up experiments leading to slightly different numbers for all hindcasts and a stronger change from 0.6 to 2.0 Natal Pulses per year in the INALT20 spin-up from 1987 -1989. However, in INALT60 only one Natal Pule occurs in the period 1987 to 1989. We corrected the numbers in Table 10 accordingly and added a reference for the method used to calculate Natal pulses. The analysis on the mechanisms behind the low number of Natal Pulses in INALT60 is beyond the scope of the current model description paper.

page 28, line 26: *The eddy-rich reference configurations range between 1.4 and 2.2 1.2 and 1.6 Natal Pulses per year in the period 2005 to 2009, that were defined as a deflection of the maximum velocity in the AC by more than two standard deviations off-shore (following Biastoch et al., 2018).*

———————————————

**Fig. R1.** 10 year mean velocities at 100 m depth in INALT10 (a) and INALT10x (b); Meridional velocities at 41.9° S (c and d; red lines in a and b), 44.1° S (e and f; blue) and 45.9° S (g and h; green).